# Evaluating the impact of Hazelwood mine fire event on students' educational development with Bayesian interrupted time-series hierarchical meta-regression

**Caroline X. Gao**[1,2,3]*, **Jonathan C. Broder**[1], **Sam Brilleman**[1], **Timothy C. H. Campbell**[4], **Emily Berger**[5], **Jillian Ikin**[1], **Catherine L. Smith**[1], **Rory Wolfe**[1], **Fay Johnston**[6], **Yuming Guo**[1], **Matthew Carroll**[4]

1 School of Public Health and Preventive Medicine, Monash University, Melbourne, Victoria, Australia, 2 Centre for Youth Mental Health, University of Melbourne, Parkview, Victoria, Australia, 3 Orygen, Parkview, Victoria, Australia, 4 Monash Rural Health, Monash University, Churchill, Victoria, Australia, 5 Department of Education, Monash University, Melbourne, Victoria, Australia, 6 Menzies Institute for Medical Research, University of Tasmania, Hobart, Tasmania, Australia

* caroline.gao@monash.edu

**Data Availability Statement:** The data underlying these analyses are owned by a third party, the Australian Curriculum, Assessment and Reporting

## Abstract

### Background

Environmental disasters such as wildfires, floods and droughts can introduce significant interruptions and trauma to impacted communities. Children and young people can be disproportionately affected with additional educational disruptions. However, evaluating the impact of disasters is challenging due to difficulties in establishing studies and recruitment post-disasters.

### Objectives

We aimed to (1) develop a Bayesian model using aggregated school-level data to evaluate the impact of environmental disasters on academic achievement and (2) evaluate the impact of the 2014 Hazelwood mine fire (a six-week fire event in Australia).

### Methods

Bayesian hierarchical meta-regression was developed to evaluate the impact of the mine fire using easily accessible aggregated school-level data from the standardised National Assessment Program-Literacy and Numeracy (NAPLAN) test. NAPLAN results and school characteristics (2008–2018) from 69 primary/secondary schools with different levels of mine fire-related smoke exposure were used to estimate the impact of the event. Using an interrupted time series design, the model estimated immediate effects and post-interruption trend differences with full Bayesian statistical inference.

Authority (ACARA). The authors do not have the authority to share or distribute the third-party data. ACARA requires users to provide formal agreement and ethical clearance to regulate the data storage and use. Other researchers wishing to access the data should follow the data access procedures outlined at https://www.acara.edu.au/contact-us/acara-data-access, which provides details of the request process and ethical approval requirements. Requests for data access need to be submitted via an online application from https://acara2.secure.force.com/datarequest/. The authors did not have any special access privileges. To ensure the reproducibility of our model, a synthetically generated data set based on the source data for the tutorial part of the paper is available at https://doi.org/10.5281/zenodo.6613107.

**Funding:** This work was funded by the Department of Health, State Government of Victoria. The paper presents the views of the authors and does not represent the views of the Department.

**Competing interests:** The authors declare that they have no known competing financial interests or personal relationships that could have appeared to influence the work reported in this paper.

## Results

Major academic interruptions across NAPLAN domains were evident in high exposure schools in the year post-mine fire (greatest interruption in Writing: 11.09 [95%CI: 3.16–18.93], lowest interruption in Reading: 8.34 [95%CI: 1.07–15.51]). The interruption was comparable to a four to a five-month delay in educational attainment and had not fully recovered after several years.

## Conclusion

Considerable academic delays were found as a result of a mine fire, highlighting the need to provide educational and community-based supports in response to future events. Importantly, this work provides a statistical method using readily available aggregated data to assess the educational impacts in response to other environmental disasters.

## Introduction

Each year, environmental disasters, such as wildfires, floods, and droughts, affect 100 to 300 million people worldwide [1]. The frequency and severity of disasters such as wildfires have increased dramatically in recent years within the context of climate change [2–4]. Children and adolescents are particularly vulnerable to experiencing difficulties in the aftermath of environmental disasters due to physiological and developmental factors, as well as their dependence on others for care, protection and decision-making, associated with their age [5, 6]. The health, social and economic disruption of disasters impose substantial impacts on children and adolescents through factors such as trauma, illness, prolonged school interruption and reduced social support [7, 8].

Exposure to disaster can result in a range of short-term and long-term psychological and neurodevelopmental consequences for children and adolescents [9, 10]. Childhood trauma exposure is known to cause emotional and behaviour issues [11, 12], ongoing sensitivity to stress [13], impair sequential development of brain structures and functions [13, 14], difficulties in attention, memory and executive function [15, 16]. Posttraumatic stress disorder (PTSD), depression, anxiety and behavioural problems are widely reported in children and adolescents impacted by environmental disasters [9, 17–19]. Wildfires have also been found to cause traumatic distress for children and adolescents [20–23]. These impacts can be carried into adulthood and, in turn, increase the risks of psychosocial problems throughout the lifespan [13, 24–26].

The psychological and neurodevelopmental impacts from trauma are important determinants of academic attainment in childhood and adolescence. The academic performance of students who have experienced trauma is generally poorer, and they are recognised to be at greater risk of early school dropout [27, 28]. Although the predominant focus of childhood trauma research has been on the impacts of maltreatment, adverse academic outcomes have been similarly observed in children and adolescents following exposure to environmental disaster [29–31].

Environmental disasters also impact children and adolescents through ongoing physical, educational, social and economic disruption. Schools may be forced to close or relocate for periods of time during and after disasters, causing interruptions and sometimes discontinuation in students' academic programs, recreational activities, and social connections [29, 32–34]. Other physical, economic and social impacts, such as damage to property, loss of job

opportunities, reduced incomes, increased distress levels and illness reported impacting adults [2, 5, 35, 36] can be potentially magnified in children and adolescents. However, the psychological and academic needs of children and adolescents can be overlooked or insufficiently responded to after disasters [20, 37–39].

Therefore, there is a need to understand the impact of environmental disasters on children and young people, particularly under the increasing threat of climate change. However, evaluating the impacts of environmental disasters on children and adolescents at a population-level is challenging, due to high costs, low response rates in surveys and lack of pre-disaster measurements necessary for evaluating changes [6, 40, 41]. Administrative data such as the national standardised academic testing results (with participant rates of about 95%) can be used to evaluate the educational impact of disasters [29]. However, accessing individual students' academic records requires parental consent and complex ethics procedures, which can be a lengthy process and particularly challenging following disaster events.

These factors can prevent a timely understanding of the degree and extent of environmental disasters' impacts, as well as how resources should be distributed to facilitate students' coping and recovery and prevent long-term deficits and inequity. Therefore, there is an urgent need for establishing advanced methods that can use readily available aggregated educational data to evaluate the impact of environmental disasters on children and adolescents.

In this study, we propose a new method to address these challenges by examining the impact of the Hazelwood mine fire. In February 2014, a bushfire ignited the Morwell coal mine adjacent to the Hazelwood Power Station in the Latrobe Valley, Victoria, Australia and burned for approximately six weeks. Whilst the flames themselves did not directly threaten homes or lives, heavy smoke concentrations throughout the six-week period, particularly in the nearby town of Morwell and the wider Latrobe valley area [42] resulted in increased mortality, physical ill-health and psychological distress in the local community [35, 43–47]. The event also caused considerable school disruption, including temporary closures and relocations [46, 48].

The Hazelwood Health Study (HHS; *www.hazelwoodhealthstudy.org.au*), established to evaluate the health and wellbeing impacts of the mine fire [46, 49], conducted a school survey to assess the psychological outcomes of the mine fire on students attending schools in affected communities. A subsequent evaluation of the National Assessment Program-Literacy and Numeracy (NAPLAN) results of survey participants suggested academic delays in highly smoke-exposed areas [50]. However, the participation rate in the school survey was relatively low, as is frequently the case in post-disaster studies [51, 52] rendering the results vulnerable to bias and requiring further evaluation. Subsequently, aggregated school-level NAPLAN data were subsequently obtained to further consolidate our findings.

In this study, we developed a Bayesian interrupted time series hierarchical meta-regression model to evaluate the impact of the Hazelwood mine fire on academic performance. Using this method, instead of individual-level data, only aggregated school-level data from standardised academic tests is required for evaluating spatial and temporal profiles of community-wide traumatic events. Here, we have presented detailed procedures and results of this novel analytic approach to inform policy and practice post- environmental disasters and assist the conduct of future research in this area.

## Methods

### Study design

All study procedures were approved by the Monash University Human Research Ethics Committee (project number: 5834) and the Victorian Department of Education and Training.

## Selection of schools and classification of exposure risks

The school-level exposures were assigned based on the Australian Statistical Geography Standard (ASGS) boundaries [53]. ASGS divides geographical areas, also known as statistical areas (SA), into four levels: SA1, the smallest census unit with a population between 200–800; SA2, comparable to a township; SA3, comparable to a local government area; and SA4, largest sub-State regions.

Due to a lack of air quality monitoring data in the initial mine fire period when pollution levels were highest [54, 55], retrospective modelling of air pollution levels was conducted by CSIRO [42]. Hourly emissions of mine fire-related particulate matter exposure ($PM_{2.5}$) were estimated using a prognostic meteorological and pollutant dispersion model at a resolution of 1km x 1km, see details published elsewhere [42]. The estimated spatial-temporal $PM_{2.5}$ concentrations were aggregated by SA2 to understand the geographical distribution of cumulative mine fire-related air pollution exposure.

The modelled mine fire-related $PM_{2.5}$ concentrations suggested that the Morwell SA2 closest to the mine fire experienced the highest smoke concentrations, which at times greatly exceeded national safety standards [42]. Prevailing winds largely restricted the dispersion of the smoke plume to the Latrobe Valley SA3 region (which includes Morwell), and that area is considered to have suffered a moderate level of exposure [42]. Further to the east, the SA3 area of Wellington Shire, with a socioeconomic profile similar to that of Latrobe Valley, had little to no exposure to the smoke. Therefore, Wellington Shire was chosen as the comparison area for this analysis. There were 69 primary and secondary schools in the region which were classified into three mine fire exposure groups: Morwell (high exposure), the remainder of the Latrobe Valley (moderate exposure) and Wellington (no/low exposure), see Fig 1.

## NAPLAN and school profile data

In Australia, NAPLAN tests are conducted annually in May for Grade 3, 5, 7, and 9 students, and NAPLAN examines educational domains of reading, writing, numeracy, spelling, and grammar/punctuation [56]. As NAPLAN assesses incremental learning, students' scores are expected to increase when they are retested at each two-year interval, from an average score of around 420 in Grade 3, to 580 in Grade 9 across individual domains [56].

Aggregated school-level NAPLAN data (mean score and SE for each domain) were requested from the Australian, Curriculum, Assessment and Reporting Authority (ACARA) for all Victorian schools between 2008 and 2018. School profiles, including total enrolments, school sector (government, non-government), school type (primary, secondary), gender proportion, and Index of Community Socio-Educational Advantage (ICSEA), were provided by ACARA for each year. ICSEA is a scale indicating socio-educational advantage of the school based on students' family background information, which ranges from about 500 (extremely disadvantaged student backgrounds) to about 1300 (extremely advantaged student backgrounds).

## Statistical methods

School characteristics (in 2014) and the NAPLAN participation rate (2008–2018) were first compared in the three exposure groups. School-level mean NAPLAN scores pre- (2008–2013) and post-mine fire (2014–2018) were visualised using box plots by exposure group for each educational domain and student grade level.

## Centring NAPLAN scores

NAPLAN tests were designed to represent the growth of students' achievement over time from Grade 3 to Grade 9, with scores expected to increase with increasing grade level. As the test

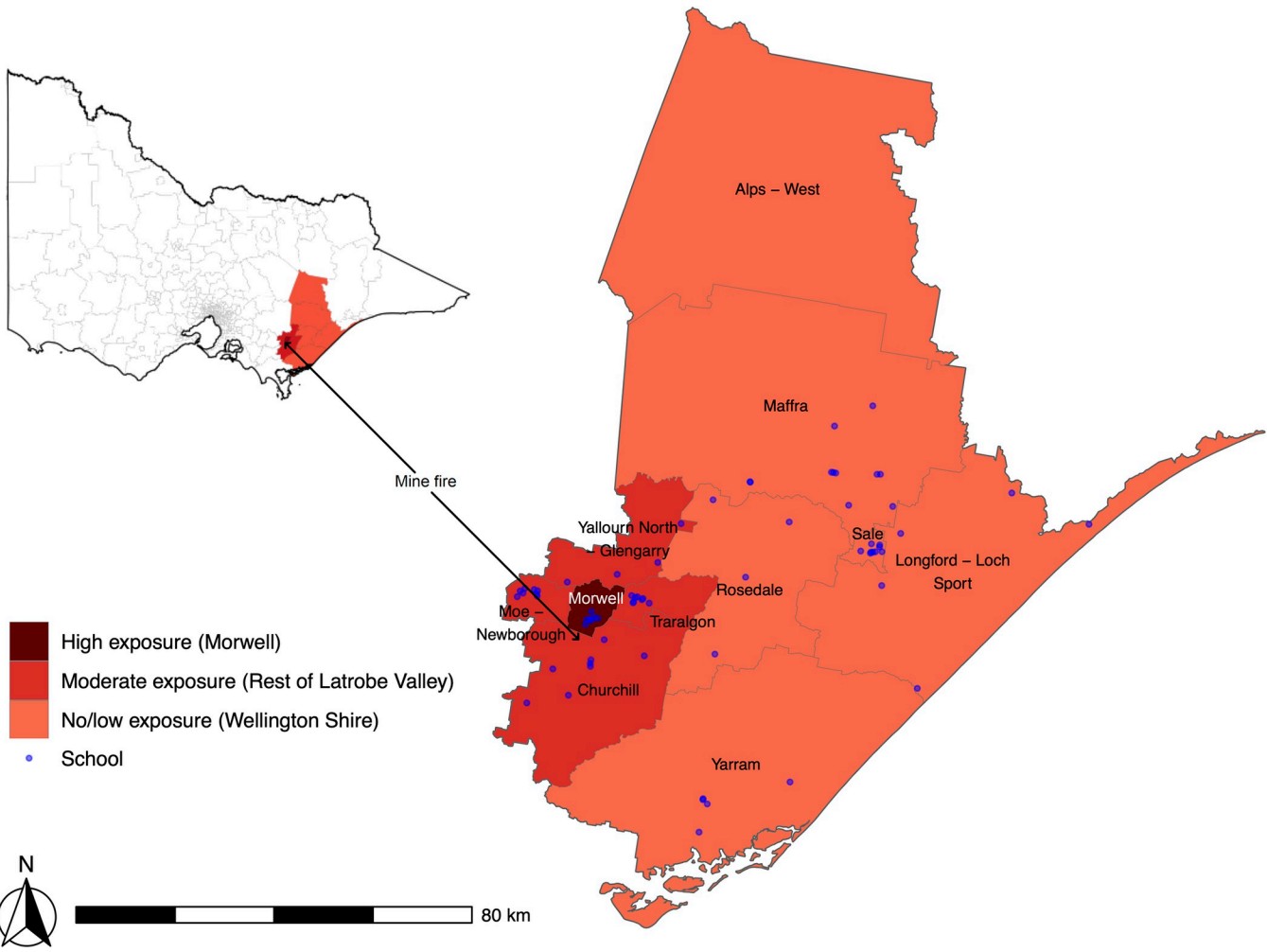

**Fig 1. Map of selected mine fire exposed SA2s in Victoria.** Superimposed dots are the location of schools in the impacted areas.

score can be interpreted in terms of academic progression, it was not standardized. However, to make results comparable across students' grades and domains, all scores were first centred against the matching mean scores in regional Victoria (for the same year, grade and educational domain).

## Bayesian interrupted time series hierarchical meta-regression

Hierarchical two-level meta-regression models were carried out in a Bayesian modelling framework to estimate the association between mine fire exposure and centred NAPLAN scores. This approach directly models distributions of students' performance within schools using mean scores and standard errors, which follows the meta-analysis approach to ensure unbiased estimates that are comparable with the individual-level model without confounder adjustment (assuming there is no ecological bias). Interrupted time series models using only means with linear regression instead of meta-regression will be inefficient (significant loss of statistical power) and biased towards smaller schools (large schools with more students should be given higher weights, e.g., using regression models weighted by the number of students or meta-regressions with inverse variance as weights). Although hierarchical meta-regressions

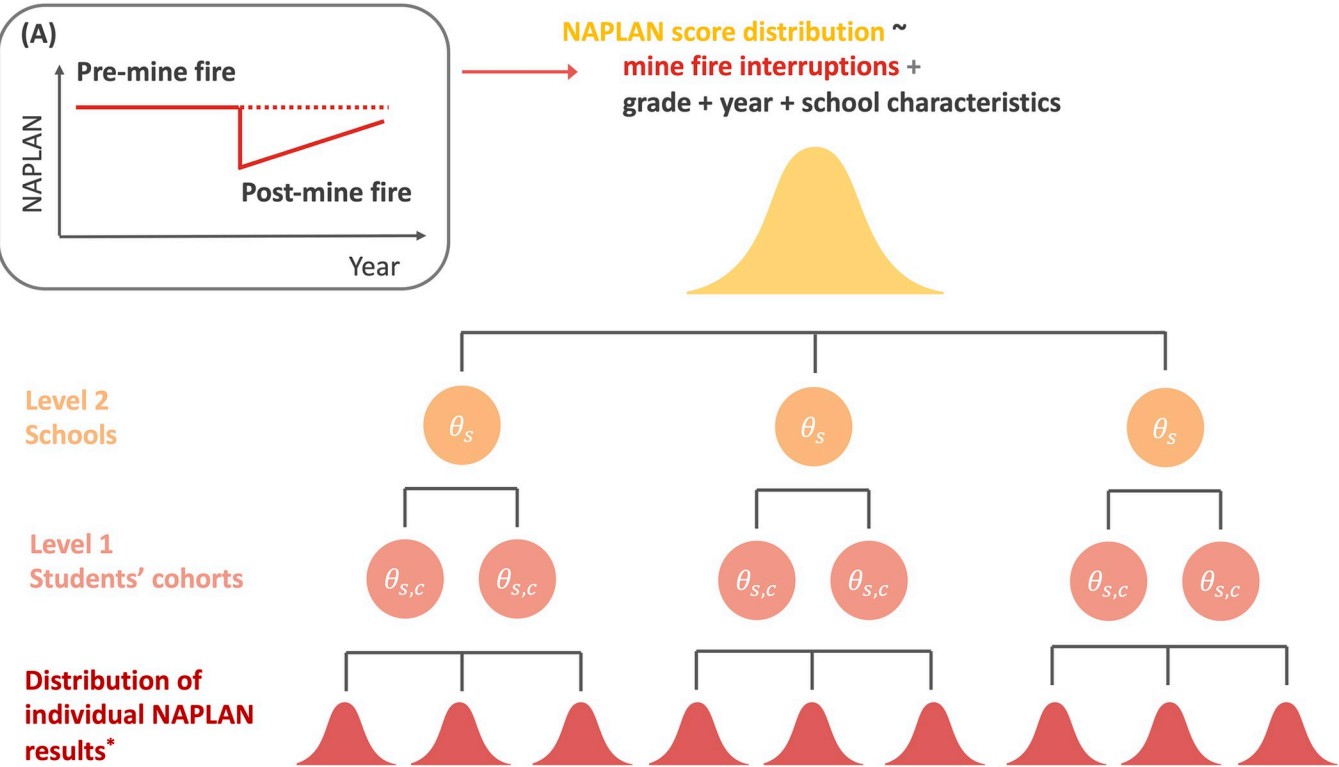

**Fig 2. Illustration of Bayesian interrupted time series hierarchical meta-regression.** *Means and SE of test results each year for individual students' grade level at each school. The subgraph (A) represents a theoretic model of the mine fire interruption effect.

can be carried out in a non-Bayesian framework, the complex model structure can introduce convergence difficulties and problematic confidence intervals [57]. In the Bayesian framework, prior information can also be incorporated.

The modelling framework is illustrated in Fig 2. The first level random effects (random intercepts) were modelled as student cohorts, e.g., the cohort of students progressing from Grade 3 in 2014 to Grade 5 in 2016 at the same school. The nested second level random effects (random intercepts) were modelled at the school-level. The centred mean NAPLAN score at Grade $g$ level for cohort $c$ students in given school $s$ was modelled as follows:

$$y_{s,g,c} \sim N(\mathbf{X_{s,g}}\beta + \theta_{s,c} + \theta_s + \theta_e, \tau_{s,g,c}^2) \tag{1}$$

The terms $\theta_s$ and $\theta_{s,c}$ are the random effects for the school $s$ and cohort $c$ in that school, respectively. $\theta_e$ is the random error term. We assume $\theta_{s,c} \sim N(0, \sigma_{s,c}^2)$, $\theta_s \sim N(0, \sigma_s^2)$ and $\theta_e \sim N(0, \sigma_e^2)$. $\tau_{s,g,c}$ is the standard error of the mean NAPLAN score of the given school, student cohort and grade level (input data). The matrix $\mathbf{X_{s,g}}$ for fixed effects includes potential confounding factors: school characteristics (ICSEA, total enrolments, percentage of girls, school sector), grade level, and the interrupted time series terms to model the mine fire exposure effect variables detailed below.

The effect of exposure to smoke during the mine fire was evaluated using an interrupted time series design (Bernal et al., 2017) with a time-specific interruption variable to capture the immediate effect of the mine fire on the NAPLAN results of that year, and a change in trend from pre- to post-mine fire (an interaction between the time interruption variable and year), see Fig 2(A). It was hypothesised that there would be no interruption effect in the no/low

exposure group. Mine fire interruption effects and post interruption trend differences were evaluated separately for the high and moderate exposure groups. The interrupted time series terms in $\mathbf{X_{s,g}}\beta$ in Eq ($1$) are summarised as follows:

$$\beta_t T + \beta_{m,\text{pre}}E_m + \beta_{m,int}E_m I_{post} + \beta_{m,trend}E_m I_{post} T_{post} + \\ \beta_{h,\text{pre}}E_h + \beta_{h,int}E_h I_{post} + \beta_{h,trend}E_h I_{post} T_{post} \tag{2}$$

Here $T$ is the time variable (0 for 2008, 1 for 2009 etc.) and coefficient $\beta_t$ represents the long-term trend in the study aa. $\beta_{m,\text{pre}}, \beta_{m,\ int}, \ldots, \beta_{h,trend}$ represent coefficients related to mine fire exposure. $E_m$ and $E_h$ are indicator variables for moderate and high exposure. $I_{post}$ is the indicator variable for pre- or post-mine fire (0 for 2008–2013 and 1 for 2014–2018). $T_{post}$ is the post-mine fire time variable (0 for 2008–2014, 1 for 2015, 2 for 2016 etc.). Therefore, $\beta_{m,\text{pre}}$ and $\beta_{h,\text{pre}}$ can be interpreted as the pre-mine fire differences when comparing moderate and high exposure schools with no/low exposure schools. These two coefficients are subsequently referred to as the fixed intercepts. The coefficients $\beta_{m,int}$ and $\beta_{h,int}$ can be interpreted as the mine fire interruptions effect (relative to the developmental trajectories of students) for moderate and high exposure schools, respectively. The coefficients $\beta_{m,trend}$ and $\beta_{h,trend}$ are the post-mine fire trend differences compared with the trend before the mine fire. Other than $\beta_{m,pre},$ $\beta_{m,int}, \ldots, \beta_{h,trend}$ the vector of coefficients, $\beta$, also contains coefficients of other confounding variables detailed above.

## Estimation and modelling framework

Estimation was conducted using Markov chain Monte Carlo (MCMC) sampling implemented in the Stan programming language [58] via the RStan package [59]. Stan is a platform for high-performance full Bayesian statistical inference using the 'No-U-Turn Sampler' (NUTS) [60]. Weakly informative prior distributions were used for standard deviations (SDs) of random effects, namely, $\sigma_{s,c} \sim TN(10, 5^2)$, $\sigma_s \sim TN(10, 5^2)$ and $\sigma_e \sim TN(10, 5^2)$, where TN is the truncated normal distribution with support over the range. A mean of 0 is normally used for prior distributions of the random effects to take more advantage of the partial pooling effect of the multi-level model [61]. In our case, the mean of 10 was chosen as the high-performance schools were slightly underrepresented in the area and variations across all regional Victorian schools were used (based on a preliminary exploratory evaluation). Weakly informative prior distributions $N(0, 50^2)$ were used for all fixed effect parameters as non-informative priors based on uniform distributions can lead to a range of model fitting issues [58, 62].

Separate models were estimated for each testing domain. Results were reported as the posterior mean of estimated coefficients (4 Monte Carlo chains with 2000 iterations each), 95% credible interval (CI) and the probability of estimated coefficients ($\beta$) being greater or less than 0. Predicted centred NAPLAN score for schools were calculated and visualised for each testing domain and exposure group using line plots.

## Sensitivity analyses

A range of sensitivity analyses were undertaken to test the robustness of results, including using different prior distributions and excluding cohort random effects. In addition, two schools in Morwell were relocated during the mine fire event and remained at their relocation sites for an extended period afterwards; hence sensitivity analyses were conducted excluding the two relocated schools. Code for fitting the models using synthetic data is provided in the Supporting Information I.

**Table 1. School profile (in the year 2014) and NAPLAN participation rate by exposure area.**

|  | Overall (N = 69) | No to low exposure (N = 34) | Moderate exposure (N = 28) | High exposure (N = 7) |
|---|---|---|---|---|
| ICSEA | 963 (938, 998) | 974 (954, 998) | 962 (942, 1,006) | 922 (906, 944) |
| Proportion of girls | 0.49 (0.45, 0.51) | 0.49 (0.45, 0.51) | 0.49 (0.46, 0.51) | 0.47 (0.44, 0.51) |
| Total enrolments | 153 (79, 316) | 108 (47, 254) | 165 (118, 340) | 204 (142, 280) |
| School type |  |  |  |  |
| Combined | 1 (1.4%) | 1 (2.9%) | 0 (0%) | 0 (0%) |
| Primary | 60 (87%) | 29 (85%) | 25 (89%) | 6 (86%) |
| Secondary | 8 (12%) | 4 (12%) | 3 (11%) | 1 (14%) |
| School sector |  |  |  |  |
| Government | 53 (77%) | 26 (76%) | 22 (79%) | 5 (71%) |
| Non-government | 16 (23%) | 8 (24%) | 6 (21%) | 2 (29%) |
| Participation rate |  |  |  |  |
| 2008 | 0.97 (0.93, 1.00) | 0.97 (0.93, 1.00) | 0.97 (0.94, 1.00) | 0.96 (0.92, 0.99) |
| 2010 | 0.94 (0.88, 0.98) | 0.97 (0.89, 1.00) | 0.94 (0.91, 0.96) | 0.82 (0.80, 0.85) |
| 2012 | 0.94 (0.91, 1.00) | 0.97 (0.92, 1.00) | 0.93 (0.90, 0.96) | 0.92 (0.88, 0.95) |
| 2014 | 0.96 (0.90, 0.98) | 0.96 (0.95, 0.98) | 0.93 (0.88, 0.97) | 0.94 (0.92, 0.96) |
| 2016 | 0.94 (0.91, 0.97) | 0.95 (0.91, 0.97) | 0.94 (0.91, 0.97) | 0.92 (0.88, 0.93) |
| 2018 | 0.94 (0.89, 0.97) | 0.94 (0.88, 0.97) | 0.94 (0.90, 0.95) | 0.97 (0.91, 0.99) |

Note: Statistics presented are either the median presented as IQR (Q1, Q3) or sample size presented as n (%)

## Results

### School characteristics

The characteristics of schools across exposure groups are presented in Table 1. The ICSEA socio-educational advantage scores were lower for schools in the high exposure group (Morwell) compared to schools in the moderate and no/low exposure groups. Other school characteristics, including the percentage of girls, the number of students, the school sector and the participation rates of the NAPLAN were comparable between exposure groups.

### NAPLAN score pre-and post-mine fire

Distributions of NAPLAN scores pre- and post-mine fire for each domain and grade level were plotted by exposure group, which are presented in Fig 3. While there was a general trend across all exposure groups for NAPLAN performance to decline post-mine fire, this was greater in higher exposure schools.

### Bayesian hierarchical meta-regression models

The results of the Bayesian hierarchical meta-regression models are presented in Table 2 and Fig 4. Compared with the Victorian regional average, there was an estimated downward trend across all domains of testing for the schools in the three exposure groups (see Fig 4 and S1-S5 Tables in S2 File). As shown in Table 2, NAPLAN scores were found to be similar between schools in the moderate and no/low exposure group pre-mine fire after controlling for other confounding factors (i.e., there was no evidence that the fixed intercept coefficients for moderate exposure differed from 0). However, pre-mine fire NAPLAN scores were estimated to be lower in schools in the high exposure group for most domains when compared with no/low exposure schools (fixed intercept for high exposure ranged between -3.68 to -13.41, Table 2).

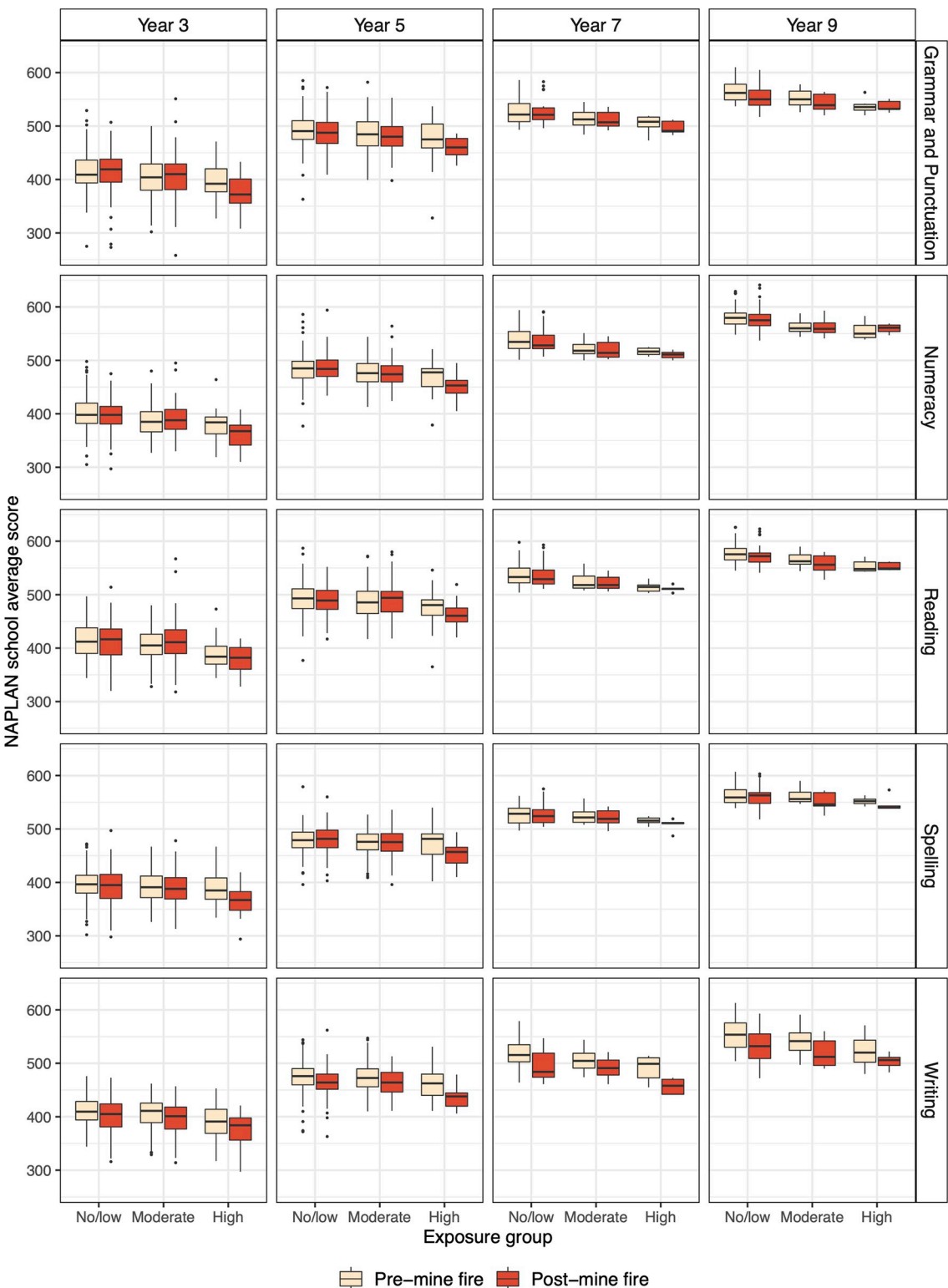

**Fig 3. Box plot of school mean NAPLAN score for pre-mine fire period and post-mine fire period by exposure group for each domain and grade level.**

**Table 2. Estimated intercept, mine fire interruption effect and post-mine fire trend difference for moderate and high exposure schools estimated from Bayesian hierarchical meta-regression models.**

|  | Moderate exposure | | | High exposure | | |
|---|---|---|---|---|---|---|
|  | $\beta$ (95%CI) | P($\beta$<0) | P($\beta$>0) | $\beta$ (95%CI) | P($\beta$<0) | P($\beta$>0) |
| **Grammar and Punctuation** |  |  |  |  |  |  |
| Fixed intercept | -3.9 (-13.13, 5.31) |  | 0.20 | -11.61 (-25.28, 2.18) |  | 0.05 |
| Mine fire interruption effect | -1.78 (-5.73, 2.13) |  | 0.19 | -10.91 (-18.68, -3.08) |  | 0.004 |
| Post-mine fire trend difference | 0.92 (-0.51, 2.35) | 0.11 |  | 2.37 (-0.73, 5.37) | 0.06 |  |
| **Numeracy** |  |  |  |  |  |  |
| Fixed intercept | -5.66 (-13.94, 2.60) |  | 0.08 | -12.07 (-24.26, 0.23) |  | 0.028 |
| Mine fire interruption effect | -0.44 (-4.20, 3.28) |  | 0.41 | -10.9 (-17.98, -3.67) |  | <0.001 |
| Post-mine fire trend difference | 0.79 (-0.56, 2.13) | 0.13 |  | 1.61 (-1.21, 4.43) | 0.13 |  |
| **Reading** |  |  |  |  |  |  |
| Fixed intercept | -1.87 (-10.19, 6.39) |  | 0.33 | -13.41 (-26.26, -0.19) |  | 0.024 |
| Mine fire interruption effect | -1.18 (-4.84, 2.57) |  | 0.27 | -8.34 (-15.51, -1.07) |  | 0.013 |
| Post-mine fire trend difference | 0.69 (-0.61, 2.01) | 0.15 |  | 1.05 (-1.63, 3.85)) | 0.23 |  |
| **Spelling** |  |  |  |  |  |  |
| Fixed intercept | -0.52 (-8.45, 7.05) |  | 0.45 | -3.68 (-16.16, 8.34) |  | 0.27 |
| Mine fire interruption effect | -1.44 (-4.91, 2.09) |  | 0.21 | -10.31 (-17.39, -3.38) |  | 0.003 |
| Post-mine fire trend difference | -0.42 (-1.68, 0.85) | 0.74 |  | -1.49 (-4.20, 1.21) | 0.86 |  |
| **Writing** |  |  |  |  |  |  |
| Fixed intercept | 1.01 (-6.83, 8.71) |  | 0.61 | -7.53 (-19.63, 4.62) |  | 0.11 |
| Mine fire interruption effect | 3.56 (-0.52, 7.55) |  | 0.96 | -11.09 (-18.93, -3.16) |  | 0.003 |
| Post-mine fire trend difference | -0.6 (-2.01, 0.82) | 0.8 |  | 2.44 (-0.60, 5.50) | 0.06 |  |

Note: All regression coefficients ($\beta$), 95% Credible Intervals (95% CI), and posterior probabilities of coefficients below or above 0 [P($\beta$<0), P($\beta$>0)] were estimated from multivariate Bayesian hierarchical meta-regression models, controlling for school-level confounders including ICSEA, total enrolments, percentage of girls, school sector, grade level, long-term. Mine fire interruption effects and post-mine fire trend differences were estimated using the interrupted time series design.

Table 2 also shows that there were substantial interruption effects post-mine fire in high exposure schools across all academic domains, with estimated mean score reductions of between 10.31 and 11.09 for grammar and punctuation, numeracy, spelling and writing, and 8.34 for reading. Typically, NAPLAN scores increase by about 27 points per year [56], which means that the delay in educational attainment in high exposure schools was equivalent to four to five months.

After the initial drop in academic performance subsequent to the mine fire, there was evidence that writing and grammar and punctuation scores began to recover (positive slope) in high exposure schools, see Fig 4. However, there was no such improvement post-mine fire for spelling in high exposure schools (estimated slope of -1.49 post-mine fire; 95%CI: -4.20, 1.21). Furthermore, the predicted centred NAPLAN scores in 2018 were found to be lower compared to the pre-mine fire period in all academic domains for high exposure schools (see Fig 4), indicating incomplete recovery five years post-mine fire.

## Sensitivity analysis

The choice of prior distributions for the SDs of random effects were found to have little impact on results; however, using a weakly informative prior distribution reduced the time taken to fit the Bayesian models compared with using a non-informative prior distribution. Sensitivity analysis excluding the cohort random effects (only the school level clustering is considered) produced very similar results except for a slightly larger mine fire interruption effect for high

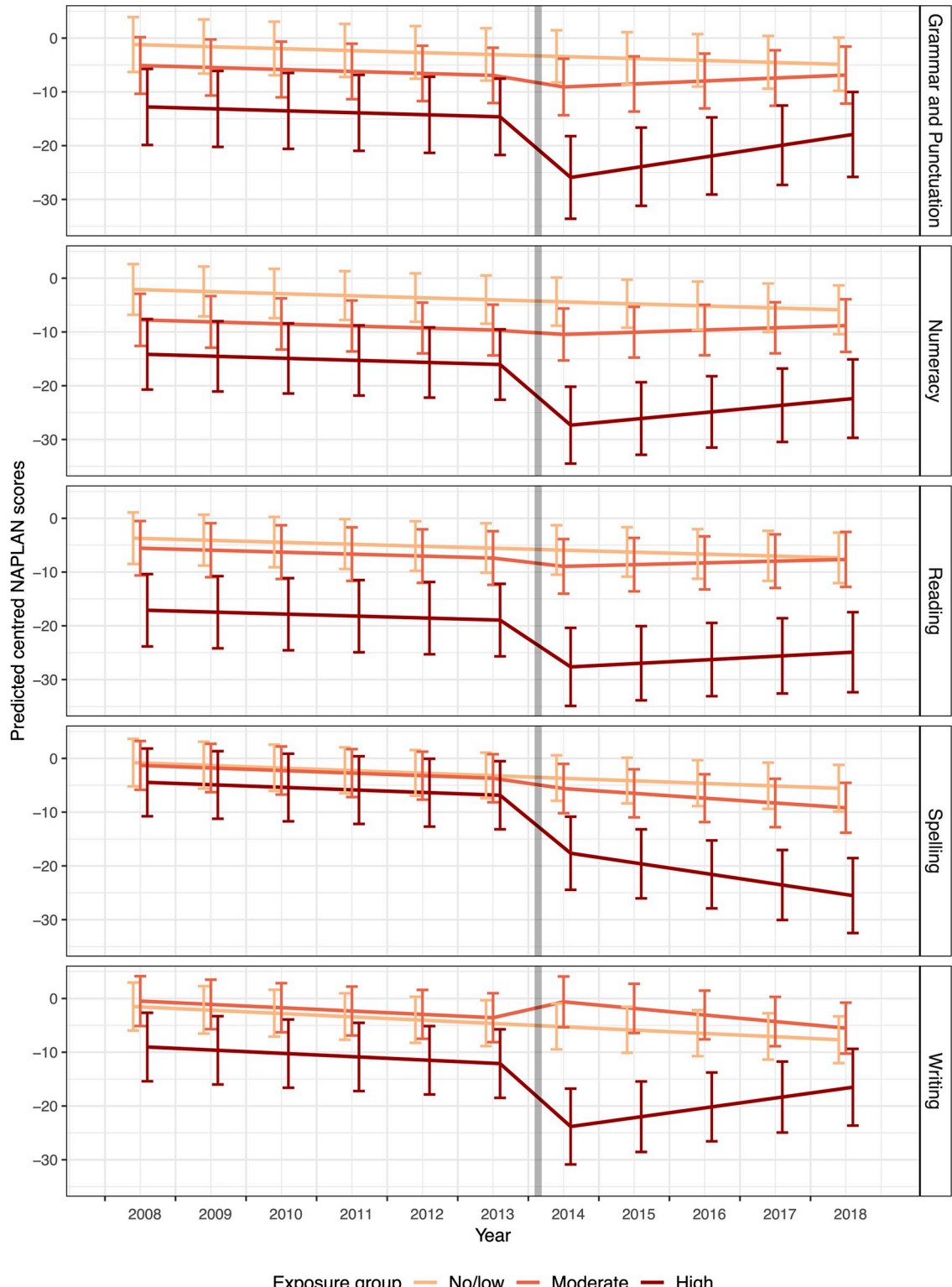

**Fig 4. Trends of predicted centred NAPLAN scores (predicted margins when all covariates are fixed at reference values) with error bars by exposure group for each academic domain.** Note: centred NAPLAN scores represent score differences of schools compared with the regional Victorian average scores in the matching year, grade level and domain; the grey line indicates the time of the mine fire.

exposure schools (see S6 Table in S2 File). Models excluding the two relocated schools showed results that were consistent with all schools, but with slightly smaller interruption effects (see S7 Table in S2 File).

## Discussion

This study presents an innovative method for evaluating the impact of disasters on students' academic performance using only readily accessible aggregated school-level data. The results suggest that the Hazelwood mine fire had a major detrimental impact on academic performance in schools located within areas highly exposed to the smoke during the event. The impact of the event on academic achievement was consistent across all NAPLAN testing domains in high exposure schools, which is comparable with a four- to five-month delay in educational attainment. While there was some recovery in academic performance in high exposure schools across all academic domains except for spelling, performance levels remained below those seen prior to the mine fire four years afterwards. Given that academic under-achievement can lead to unemployment, social and economic disadvantage, and ill-health later in life [63], it is critical that the impacts of climate disaster on the academic progression of students are recognised and remedied.

### Links between the mine-fire and academic performance

The impact of the mine fire on academic performance may be attributable to a combination of factors. The mine fire led to ongoing distress experienced by students, teachers and parents in the smoke affected community post the mine fire [35, 48, 50, 64]. Hazelwood mine fire-related air pollution has also been linked to a range of adverse physical health effects such as respiratory morbidity [44, 45, 47, 65]. More broadly, the communities' wellbeing was also negatively affected, with local residents reporting a loss of trust in authorities and feeling abandoned with little support provided [66]. The subsequent closure of the Morwell mine and Hazelwood power station some three years after the fire occurred introduced additional social and economic challenges in an already disadvantaged community [67, 68]. These factors may have impacted students' learning abilities both directly and indirectly.

The mine fire also introduced disruption to day-to-day operations in nearby schools [48]. Two schools were relocated causing direct educational interruptions as well as threats to students' sense of safety and security [48]. The models excluding these two schools showed a slightly lower level of interruption in high exposure schools compared with when they were included. This suggests that school relocation might have had an adverse impact on academic performance in addition to the mine fire exposure. Prior research has found that school relocation has a negative effect on students as well as teachers. Students are impacted by the change of routine, additional time travelling to new school locations, and uncertainty about the permanency of school relocation. Teachers experience similar uncertainty and concern about the effects of school relocation on students, and have reported that school relocation increases their workload and decreases time spent in the classroom with children [48].

Fire incidents are common in coal mines [69] and, all too often, mining project impact assessments adopt a narrow range of health and wellbeing domains that omit consideration of epidemiological evidence and potential social consequences [70, 71]. Accordingly, the impacts that events such as the Hazelwood mine fire can have on academic outcomes among school-aged young people warrant consideration within mine project planning and impact assessment, more generally, within disaster risk reduction planning and education department policies.

The potential links between air pollution and cognitive development, educational and behavioural outcomes in children is a rapidly growing area of research. Associations have been found between chronic ambient air pollution and cognitive function, as well as educational attainment [72–74]. A recent study suggested a link between academic underachievement, particularly in spelling, and prenatal airborne polycyclic aromatic hydrocarbons (PAH) exposure [75], a pollutant also found in the Hazelwood mine fire smoke [76]. However, we are not aware of previous research evaluating associations between medium-duration air pollution events and long-term academic outcomes. Although our study has shown a clear impact of the mine fire on academic performance, the model cannot delineate between the potential underlying mechanisms related to the exposure, such as psychological trauma, disruption to schooling, or air pollution exposure. Accordingly, further investigation is needed to understand the relative contributions of different mechanisms by which disaster exposure impacts academic performance, which could inform the development of targeted interventions.

## Improvements post-mine fire

Gradual improvements in academic performance were observed across most domains post-mine fire among students from high exposure schools. This may be the result of a slow recovery at both the individual and the community level. Increases in school funding from the Victorian Government to upgrade school facilities and a range of implemented government strategies focused on improving health, wellbeing and access to services in the impacted community, may have assisted the recovery process [77, 78]. Students' performance in the spelling domain continued to deteriorate post-mine fire, which may be related to idiosyncrasies in the English language orthography [79], or difficulties in teaching spelling [80, 81]. As difficulties in spelling may persist and lead to arrested writing development subsequently [82], further investigation is needed.

## Comparison with other studies

Although a theoretical link between disasters and educational outcomes has been well-established [83], most studies have been limited to evaluations of how disasters impacted school attendance or dropout rates [30, 31, 84], and academic delays have rarely been investigated. One study suggested more than 75% of the African American children evacuated in response to Hurricane Katrina (which resulted in the loss of more than 1,300 lives, 800,000 homes and 110 schools) experienced a decline in grades [34]. To our knowledge, only Gibbs and colleagues have previously evaluated the impact of disasters using national standardised academic performance tests. Their study found that exposure to the 2009 Black Saturday bushfire in Australia (loss of 173 lives, 2,000 homes and 3 schools) was associated with delays in academic achievement from Grade 3 to 5 in reading and numeracy but not in writing, spelling, and grammar domains [29]. However, only academic progressions of a single cohort of students (who were in Grade 3 in 2011 and were around 7 years old at the time of the event) were evaluated. Our study, on the other hand, has utilized a time series design to assess academic achievement across all grade-levels completing NAPLAN testing and incorporated more historical data, which sheds further light on academic impacts across age groups.

Longer-term social outcomes after man-made or industrial disasters have the potential to be worse than in cases of natural disasters [85], however, it is at present unclear whether this also applies to academic outcomes. Furthermore, while there is little research available for assessing the relative impacts of different types of disasters on academic outcomes, it is reasonable to anticipate that more severe disasters will have a potentially greater impact on academic outcomes. For instance. if school infrastructure was destroyed, or if students were exposed to the trauma of having lost family or classmates in the event, then academic consequences might

be expected to be worse than in the case of the Hazelwood mine fire. Accordingly, seeking to compare how different types of disaster shape academic outcomes would be a useful direction for future research.

## Statistical modelling approach

To the best of our knowledge, this is the first study using a Bayesian meta-regression method to evaluate aggregated school-level academic outcomes. Existing studies evaluating the impact of environmental factors on academic outcomes mostly used individual-level data [29, 86, 87]. Where aggregate data was used, it was limited to school-level mean scores or proportions [88–90], which ignores the variations in score distributions.

Our method has several strengths. Firstly, it enables the evaluation of the spatial (geographical regions of no/low exposure to high exposure areas) and temporal profile of the impact, which could be used to inform policy and resource allocation in academic settings after disasters. The inclusion of an appropriate control group of schools from areas with similar socio-economic circumstances, but not exposed to smoke during the fire, countered the problem of potential time-varying confounders (e.g., changes in tests, education policy or other major interruptions) and enabled the impact of the mine fire on academic progress in the region to be clearly isolated [91].

Secondly, the method can be applied using only readily available aggregate data at the school-level to obtain unbiased estimates without losing statistical power to detect meaningful differences. Although only 69 schools were included in our analysis, the study was effectively powered by over 30,000 students who studied in these schools from 2008 to 2018. Accessing aggregated data is a particularly useful approach to conducting research in this area as it circumvents the challenges of attempting to recruit individuals in communities' post-disaster who are already burdened and traumatized as a result of their experiences and is a much less resource-intensive approach. A similar approach can also be used without the interrupted time series design to study the associations between students' education performance and other environmental factors such as air pollution.

Lastly, the interrupted time series design facilitates the understanding of immediate interruptions as well as changes in long-term trends post-event. Compared with simple pre- and post-comparisons, this approach provides information for teachers, schools and education departments not only on the magnitude of the impact but also on the effect of post-disaster interventions to accommodate students' additional needs.

Similar models can be fitted via a frequentist regime using mixed-effect meta-regression models (e.g., estimated using restricted maximum likelihood), which can yield similar results to Bayesian models [92]. However, fitting models with the Bayesian approach has additional advantages including more stable and generalisable estimates, integration of existing knowledge using prior distributions, flexible compressions using posterior distributions and avoiding convergence and estimation problems of maximum likelihood methods [61, 93]. The choice of prior distributions did not impact our results, however, when there is high model complexity (e.g., higher numbers of potential confounding factors), penalising Bayesian models can be used with shrinkage prior distributions [94, 95].

## Limitations and future direction

This paper provides an easily adaptable method to evaluate the impact of different types of community-level traumatic exposures (e.g., disasters and disease outbreaks) on students' academic progression using only aggregated school-level data. However, the study has recognised limitations. The implemented model assumes that student NAPLAN scores within schools

were normally distributed, which could be unrealistic if, for example, distributions were skewed or the number of students is small. Although the number of schools in the high exposure group was modest, most of these schools had over 150 students enrolled, so the mine fire interruption effects identified are likely to be robust. More complicated models such as random slopes, non-linear long-term trends and autocorrelation structure, were not considered due to the unwarranted increase in modelling complexity with limited data. These models can be applied when there are more schools and time points available. More detailed region-specific data on other risk factors, such as service availability, were not available for this analysis, but could in principle be included in the proposed model.

Outcomes across multiple learning domains were compared which can introduce challenges for interpretation. In this study, it is likely that interruptions caused by the mine fire to school routines were distributed similarly across all learning domains; however, the level of impact of this disruption on students' subsequent progression within each learning domain may differ. Therefore, each domain was evaluated separately to investigate for any differing impacts and trends between domains. Alternatively, if there was an assumption or interest in evaluating a consistent impact across learning domains, all could be included in one multilevel model with an additional nested level [61].

Furthermore, the aggregated nature of the data precluded controlling for individual-level risk factors and, hence, there is potential for ecological bias within the study. While the interrupted time series design was used to allow post-mine fire scores in individual schools to be compared relative to their past, there may be changes in students' profiles over time, which may impact our findings. For example, some families may have left the region due to the subsequent closure of the mine and power station. Advanced population-level data linkage methods can be used to explore these issues. However, as noted above, accessing population-level data can be challenging and time-consuming.

## Conclusions

This study provides evidence indicating that an extended air pollution event, the Hazelwood mine fire, resulted in a delay in academic performance across multiple educational domains, which had not fully recovered several years after the event occurred. While most research to date has focused on the educational impact of maltreatment, or of major disasters causing significant loss of property and life, the current study shows that a community-wide traumatic event posing a minimal immediate risk to life and property can also have considerable long-term educational impacts. This finding highlights the substantial vulnerability of children and adolescents and the need to respond to community-wide disaster events, providing targeted support during and following the event, in the hope of preventing or ameliorating any educational impacts.

This paper provides a novel statistical method for using readily available aggregated data to assess the educational impacts of disasters. Implementing research programs post-disaster is enormously challenging. Accordingly, an approach that enables accurate and timely assessment of educational impacts without impost on the community is invaluable. This model could be applied to investigate the effect of other extended events on academic achievement, for instance, the COVID-19 pandemic, which has impacted students' access to, and delivery of, schooling worldwide.

## Supporting information

**S1 File. Tutorial for Bayesian interrupted time series hierarchical meta-regression.**
(PDF)

**S2 File. Supplementary tables.**
(PDF)

## Acknowledgments

We wish to thank the Latrobe Valley and Gippsland communities for their support and participation in the Hazelwood Health Study. We also like to acknowledge Prof Rob Hyndman for his generous sharing of the Rmarkdown LaTex template for writing this paper in the Rmarkdown environment.

## Author Contributions

**Conceptualization:** Caroline X. Gao, Matthew Carroll.

**Data curation:** Caroline X. Gao, Jonathan C. Broder, Timothy C. H. Campbell, Matthew Carroll.

**Formal analysis:** Caroline X. Gao, Jonathan C. Broder, Sam Brilleman.

**Funding acquisition:** Jillian Ikin, Fay Johnston, Yuming Guo, Matthew Carroll.

**Methodology:** Caroline X. Gao, Sam Brilleman, Emily Berger, Jillian Ikin, Rory Wolfe, Fay Johnston, Yuming Guo, Matthew Carroll.

**Project administration:** Timothy C. H. Campbell, Jillian Ikin, Matthew Carroll.

**Supervision:** Rory Wolfe.

**Validation:** Caroline X. Gao, Jonathan C. Broder, Sam Brilleman, Catherine L. Smith.

**Visualization:** Caroline X. Gao.

**Writing – original draft:** Caroline X. Gao.

**Writing – review & editing:** Caroline X. Gao, Jonathan C. Broder, Sam Brilleman, Timothy C. H. Campbell, Emily Berger, Jillian Ikin, Catherine L. Smith, Rory Wolfe, Fay Johnston, Yuming Guo, Matthew Carroll.

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
