## [Decision Letter · Decision Letter 0]

11 May 2022

PONE-D-21-37171

Evaluating the impact of Hazelwood mine fire event on students’ educational development with Bayesian interrupted time-series hierarchical meta-regression

PLOS ONE

Dear Dr. Gao,

Thank you for submitting your manuscript to PLOS ONE. After careful consideration, we feel that it has merit but does not fully meet PLOS ONE’s publication criteria as it currently stands. Therefore, we invite you to submit a revised version of the manuscript that addresses the points raised during the review process.

We apologise for the delay in the peer review process.  At the moment it is difficult to find reviewers (and editors) in general and even more difficult for this particular manuscript.  You will find comments from myself and a reviewer below which will need to be addressed in sufficient detail.  Further delays in the peer review process are not expected if you are able to provide a response within the allocated time period.

We look forward to receiving your revised manuscript.

Kind regards,

Darren Wraith

Academic Editor

PLOS ONE

**Journal requirements:**

“This work was funded by the Victorian Department of Health and Human Services.”

“NO authors have competing interests.”

5. We noted in your submission details that a portion of your manuscript may have been presented or published elsewhere. [The manuscript is published on the preprint server https://www.medrxiv.org/content/10.1101/2021.03.28.21254516v2] Please clarify whether this [conference proceeding or publication] was peer-reviewed and formally published. If this work was previously peer-reviewed and published, in the cover letter please provide the reason that this work does not constitute dual publication and should be included in the current manuscript.

7. We note that you have indicated that data from this study are available upon request. PLOS only allows data to be available upon request if there are legal or ethical restrictions on sharing data publicly. For more information on unacceptable data access restrictions, please see http://journals.plos.org/plosone/s/data-availability#loc-unacceptable-data-access-restrictions.

8.  Your ethics statement should only appear in the Methods section of your manuscript. If your ethics statement is written in any section besides the Methods, please delete it from any other section.

**Additional Editor Comments:**

Apologies for the delay with the peer review. As mentioned it has been difficult to find suitable reviewers for this manuscript. You will find comments from one reviewer which are reasonable and should be addressed in detail. In addition, I have some comments:

- At times the application is referred to as an example of the method being used more generally in climate change work but there is little discussion or comparison of alternative models available for this. As the authors will be aware, there has been some recent literature on interrupted time series design and comparisons made between different approaches (see Turner et al, 2021 as one example). Also, the model used is a fairly simplified version of a time series model in general without seasonality, cyclical effects and autocorrelation being used. This is perhaps fine for this setting, but again if it is being proposed as a method then the simpler case being used could be clearly defined from the general method (and limitations highlighted).

- The factors responsible for the decline in academic achievement are not convincingly discussed given the countervailing forces involved. For example, is there any information available about levels of funding to support the schools affected and specific interventions that were made? The results then seem to be in spite of these interventions being available (and perhaps were not effective in some areas?). Forces influencing the quality of teaching don’t seem to be discussed but could relate to teachers leaving the area, staff turnover, trauma experienced by the teachers, etc. Other factors not measured/available in the model also could be discussed in terms of changes to curriculum/schooling, any other events in the area around this time which could be involved. Some of this could be speculative but gives some insight into the factors affecting the quality of education for schools in particular areas. This could also be part of the translation public health/educational messaging in the Discussion section which is reasonably weak at the moment in terms of this.

- The influence of the two relocated schools seems to be important in terms of the results (Table S7) but the discussion on this seems to be only a sentence in the sensitivity analysis. Thus, there is room to discuss the influence of this with more insight as to the overall results. This is important given the Discussion on possible reasons for the decline in scores.

- I couldn’t see a clear definition for what was considered pre and post time periods in terms of NAPLAN scores (e.g when NAPLAN was carried out). If it is there make it clearer. Relating to this is the sensitivity of the results in terms of the timing of NAPLAN results and definitions of pre and post mine periods? NAPLAN results may not an immediate indicator for that year but rather of previous learning so there could/will be a time delay involved.

Reviewers' comments:

Reviewer's Responses to Questions

**Comments to the Author**

1. Is the manuscript technically sound, and do the data support the conclusions?

Reviewer #1: Yes

2. Has the statistical analysis been performed appropriately and rigorously? 

Reviewer #1: Yes

3. Have the authors made all data underlying the findings in their manuscript fully available?

Reviewer #1: Yes

4. Is the manuscript presented in an intelligible fashion and written in standard English?

Reviewer #1: Yes

5. Review Comments to the Author

Reviewer #1: The authors present a method to use available aggregated data on students performance to estimate the impact of a significant environment event (mine fire), applying Bayesian hierarchical meta-regressions and an interrupted time series design. The study is well conducted, its methodology is well presented and supported, and the results are consistent with some previous research on the topic. The study is an important contribution to the scientific community and the manuscript can be published after addressing the aspects described below:

Abstract

Line 30. “With the increasing threat of climate change” this does not link to what is provided in the methods and results.

Line 31. “a new analytical framework to evaluate the impact of climate disasters”. This is a very comprehensive (and ambitious) objective. Also, this was not addressed this way in the methods, results and discussion sections.

Line 34. The word “only” can be confusing since the models include several co-variates. The authors might find a better way to highlight the advantage of the proposed methodology to use this kind of easily available data.

Line 44. Conclusion or discussion ?

Introduction

Multiple citations are from old studies. Strongly suggest considering more recent references

Lines 53, 57, 59. Suggest having more recent citations

Line 65. Consider more recent research to support the background/introduction

Parag. Starting in line 66. Some redundancy with previous paragraph. Maybe the 2 parag. can be merged.

Line 85. This statement must be supported by more recent research

Paragraphs from line 100-114 are somewhat disconnected from the last statement in line 99 that should link to the 2nd and 3rd sentences in the last paragraph of the introduction. This is the focus of the paper while the Hazelwood event is used as a case study.

Methods

Line 124-125. You can remove “the town”

Lines 127-131. Suggest some pre-explanation of the area levels (it’s a bit confusing. Not all readers would be familiar with these Australian areas).

First para. Further explanation needed to understand why and how the modelled PM25 was used to specify the selection of schools.

Line 132. Not sure of the meaning (need) of the last part of this sentence

Line 151. Some explanation on the interpretation of the ICSEA is needed (higher or lower scores indicate more SE advantage?)

Figure 1 would be benefited if the location of the fire is shown (or another figure so the spatial setting is better contextualised regarding the exposure source). Why areas west of Morwell are not considered? (maybe this links to the need of better explanation of the PM25 modelling)

Line 169: unbiased

Line 170. Any previous research using this approach?

Good inclusion of figure 2

Statistical analysis is well introduced and supported

Line 228. Some description of the priors tested in the sensitivity analyses and the main outcomes of excluding cohort RE should be included in the supp. Material

Results

Table 1 presents 2 different measures in parentheses -this needs to be better clarified in the table title.

Line 262. Review these figures

Lines 264-27 data already shown in table 2

Paragraph starting in line 290. This answers one of my comments above.

Discussion

Line 303. “The results suggest..”

Line 305. Suggest” “the detrimental impact…” (avoid confusion —impacts can also be positive)

Line 326. “The potential links…”

Line 331. “the Hazelwood mine fire smoke”

Line 363. “also” seems redundant here

Line 366. Previous section did not clarify specifically how the analysis address the spatial profile of the impacts (models do not include a spatial term?)

The discussion needs a section to discuss the method used in this study—the strengths section somewhat addresses this but there is not a formal discussion of the specifics of the methods and comparison with other research

The limitations section should address potential ecological bias and how this might affect the results and their interpretation. Also how the limitations already commented in this section might impact the findings and how this could be controlled in further research

Conclusions

Suggest naming the fire event specifically in the first line

Last line would be better commented in the discussion not in this section

6. PLOS authors have the option to publish the peer review history of their article (what does this mean?). If published, this will include your full peer review and any attached files.

Reviewer #1: No

---

## [Author Response · Author response to Decision Letter 0]

20 Jul 2022

Responses to Editor and Reviewer

Editor comments

Apologies for the delay with the peer review. As mentioned it has been difficult to find suitable reviewers for this manuscript. You will find comments from one reviewer which are reasonable and should be addressed in detail. In addition, I have some comments:

Response: Thank you for your efforts, we understand the challenges of finding reviewers for a statistical paper, particularly given the impacts of the pandemic. We have addressed the comments in relation to the paper below.

At times the application is referred to as an example of the method being used more generally in climate change work but there is little discussion or comparison of alternative models available for this. As the authors will be aware, there has been some recent literature on interrupted time series design and comparisons made between different approaches (see Turner et al, 2021 as one example). Also, the model used is a fairly simplified version of a time series model in general without seasonality, cyclical effects and autocorrelation being used. This is perhaps fine for this setting, but again if it is being proposed as a method then the simpler case being used could be clearly defined from the general method (and limitations highlighted).

Response: Our aim was to develop models to evaluate the impact of climate disasters resulting in major disruptions (see lines 92-100). Models evaluating climate change impacts that are gradual (e.g., increases in temperature and air pollution level) were not considered here. We have removed the climate change statement in the abstract as suggested by the reviewer. 

The primary method of studying the impact of climate disasters in the research literature is still via individual-level surveys. There is a merging area focusing on using secondary data [1, 2]. However, when only aggregated data is available, the time series design is perhaps more accurate compared with other methods such as simple pre and post-comparison or difference-in-difference. 

As the editor has indicated, there are many methods to estimate an interrupted time series model. A series of papers by our Monash University colleagues Simon Turner and others have provided an excellent summary and evaluation of existing interrupted time series methods [3, 4]. However, their work primarily focused on interrupted time series models for one outcome in individual series. Our study, on the other hand, has two outcome parameters (mean and SE) and multiple series, so these methods are not directly comparable. It is possible to estimate our model using a linear fixed-effects or mixed-effects model using a frequentist regime, which we are testing at the moment with a large number of schools included, in which the Bayesian models can be very costly to estimate. We have included this in the discussion part, see lines 347-439. 

“Similar models can be fitted via a frequentist regime using mixed-effect meta-regression models (e.g., estimated using restricted maximum likelihood). Although frequentist models do not take advantage of prior distributions, they generally yield similar results to Bayesian models [5]. “

The NAPLAN test is only conducted once a year at a fixed time (mid-may), so the seasonality and cyclical effects are largely not relevant in this setting (we have made this clear in the manuscript). The autocorrelation is potentially an issue in modelling NAPLAN scores, however, it is theoretically unlikely because the students being assessed in two consecutive years are different (NAPLAN tests are only conducted in grades 3, 5,7 and 9, not every grade). The same group of students were assessed every two years, which is modelled by the random intercepts of waves of students. The main limitations of our model are that random slopes and non-linear long-term trends are not modelled. As there were only 69 schools included, more complicated models may not be generalisable even if the model can be fitted via the sampling framework. These were discussed in lines 4654-469.

“More complicated models such as random slopes, non-linear long-term trends and autocorrelation structure, were not considered due to the unwarranted increase in modelling complexity with limited data. These models can be applied when there are more schools and time points available.”

The factors responsible for the decline in academic achievement are not convincingly discussed given the countervailing forces involved. For example, is there any information available about levels of funding to support the schools affected and specific interventions that were made? The results then seem to be in spite of these interventions being available (and perhaps were not effective in some areas?). Forces influencing the quality of teaching don’t seem to be discussed but could relate to teachers leaving the area, staff turnover, trauma experienced by the teachers, etc. Other factors not measured/available in the model also could be discussed in terms of changes to curriculum/schooling, any other events in the area around this time which could be involved. Some of this could be speculative but gives some insight into the factors affecting the quality of education for schools in particular areas. This could also be part of the translation public health/educational messaging in the Discussion section which is reasonably weak at the moment in terms of this.

Response: We have updated our discussion to include more details as suggested, e.g., lines 362-373, lines 385-394. The psychological trauma for both students and teachers and interruptions to school functioning are believed to be the primary driving force behind the decline. There was no immediate funding available post the mine fire, however, additional funding was allocated in later years to upgrade school facilities. While we are anecdotally aware of post-disaster programs being put in place following the mine fire, we have been unable to access any relevant information regarding this. Curriculum/schooling remains largely unchanged in the area.

The influence of the two relocated schools seems to be important in terms of the results (Table S7) but the discussion on this seems to be only a sentence in the sensitivity analysis. Thus, there is room to discuss the influence of this with more insight as to the overall results. This is important given the Discussion on possible reasons for the decline in scores.

Response: Thank you for pointing this out. We agree that these can be important for policy guidance. We have included additional discussion around this point, see lines 362-373.

“The mine fire also introduced disruption to day-to-day operations in nearby schools [6]. Two schools were relocated causing direct educational interruptions as well as threats to students’ sense of safety and security [6]. The models excluding these two schools showed a slightly lower level of interruption in high exposure schools compared with when they were included. This suggests that school relocation might have had an adverse impact on academic performance in addition to the mine fire exposure. Prior research has found that school relocation has a negative effect on students as well as teachers. Students are impacted by the change of routine, additional time traveling to new school locations, and uncertainty about the permanency of school relocation. Teachers experience similar uncertainty and concern about the effects of school relocation on students, and have reported that school relocation increases their workload and decreases time spent in the classroom with children [6]. These impacts should be considered during disaster risk reduction planning and policies of schools.”

I couldn’t see a clear definition for what was considered pre and post time periods in terms of NAPLAN scores (e.g when NAPLAN was carried out). If it is there make it clearer. Relating to this is the sensitivity of the results in terms of the timing of NAPLAN results and definitions of pre and post mine periods? NAPLAN results may not an immediate indicator for that year but rather of previous learning so there could/will be a time delay involved.

Response: NAPLAN is held annually in May, and this was previously described in line 142 (now line 174). We collected NAPLAN data for the years 2008-2018 as previously shown in line 149 (now 181). We have now added information to define the pre- (2008-2013) and post-mine fire period (2014-2018) in lines 190-191. NAPLAN testing is designed to track the academic progression of students over time. There are a few factors relating to the mine fire event which may have impacted NAPLAN results: (1) trauma and stress directly impacting students’ cognitive ability which can be immediately observed, (2) interruptions in the delivery of educational activities which can also be observed in the NAPLAN test 3 months after the mine fire, (3) long-term changes for the socioeconomic status that can be observed in a few years following the mine fire. From the results, it seems that most schools were able to gradually recover, therefore it is likely that the lag effect is not strong. 

Reviewers 1 comments:

The authors present a method to use available aggregated data on students performance to estimate the impact of a significant environment event (mine fire), applying Bayesian hierarchical meta-regressions and an interrupted time series design. The study is well conducted, its methodology is well presented and supported, and the results are consistent with some previous research on the topic. The study is an important contribution to the scientific community and the manuscript can be published after addressing the aspects described below:

Response: Thank you for these positive comments. We have addressed the comments in relation to the paper below.

Abstract: Line 30. “With the increasing threat of climate change” this does not link to what is provided in the methods and results.

Response: Thank you for alerting us to this. This text has been removed. 

Line 31. “a new analytical framework to evaluate the impact of climate disasters”. This is a very comprehensive (and ambitious) objective. Also, this was not addressed this way in the methods, results and discussion sections.

Response: We have changed this to “develop a Bayesian model using aggregated school-level data to evaluate the impact of climate disasters on academic achievement”, now lines 30-31.

Line 34. The word “only” can be confusing since the models include several co-variates. The authors might find a better way to highlight the advantage of the proposed methodology to use this kind of easily available data.

Response: The term “only” has been replaced with “easily accessible”, now line 35

Line 44. Conclusion or discussion?

Response: This heading has been changed to conclusion, now line 45. 

Introduction: Multiple citations are from old studies. Strongly suggest considering more recent references. Lines 53, 57, 59. Suggest having more recent citations. Line 65. Consider more recent research to support the background/introduction

Response: We have updated most of the references as suggested, except for a few seminal papers (e.g., Norris et al., 2002 and McFarlane et al., 1987). 

Parag. Starting in line 66. Some redundancy with previous paragraph. Maybe the 2 parag. can be merged.

Response: We agree with this suggestion. These paragraphs have now been merged, see lines 67-76.

Line 85. This statement must be supported by more recent research

Response: We have modified the sentence and included more recent publications, see lines 89-91.

“However, the psychological and academic needs of children and adolescents can be overlooked or insufficiently responded to after disasters [7-10].” 

Paragraphs from line 100-114 are somewhat disconnected from the last statement in line 99 that should link to the 2nd and 3rd sentences in the last paragraph of the introduction. This is the focus of the paper while the Hazelwood event is used as a case study.

Response: We agree that a linking statement is needed. We have included the following additional sentence at the start of the previous paragraph, now lines 106-107. 

“In this study, we proposed a new method to address these challenges by examining the impact of the Hazelwood mine fire.”

Methods Line 124-125. You can remove “the town”

Response: Thank you, this has now been removed.

Lines 127-131. Suggest some pre-explanation of the area levels (it’s a bit confusing. Not all readers would be familiar with these Australian areas).

Response: The following information has now been provided in the manuscript, lines 134-138. 

“The school-level exposures were assigned based on Australian Statistical Geography Standard (ASGS) boundaries [11]. ASGS divides geographical areas, also known as statistical areas (SA), into four levels, including: SA1 - the smallest census unit with a population between 200-800; SA2 - comparable to township; SA3 comparable to a local government area; and SA4 -largest sub-State regions.”

First para. Further explanation needed to understand why and how the modelled PM25 was used to specify the selection of schools.

Response: The following information on how the PM2.5 data were modelled and processed has now been provided, lines 139-145. 

“Due to a lack of air quality monitoring data in the initial mine fire period when pollution levels were highest [53, 54], retrospective modelling of air pollution levels was conducted by CSIRO [55]. Hourly emissions of mine fire-related particulate matter exposure (PM2.5) were estimated using a prognostic meteorological and pollutant dispersion model at a resolution of 1km x 1km, see details published elsewhere [55]. The estimated spatial-temporal PM2.5 concentrations were aggregated by SA2 to understand the geographical distribution of cumulative mine fire-related air pollution exposure.”

Line 132. Not sure of the meaning (need) of the last part of this sentence. 

Response: The air pollution model provides a high-resolution spatial and temporal estimate of mine fire PM2.5 exposure, so the exposure at the location of each school can be estimated (either using exposure level in the 1km x 1km grids estimated directly from the air pollution model or the aggregated SA2-level exposure). However, the student also had exposure at their residential home address which may or may not be close to their schools, and which we were not able to estimate given we were restricted to aggregate data. So, we divided schools into three exposure groups (high, moderate and no/low exposure) rather than using the estimated PM2.5 level at the individual school location as a continuous exposure variable. We have modified this paragraph to make it clear. 

“The modelled mine fire-related PM2.5 concentrations suggested that the Morwell SA2 closest to the mine fire experienced the highest smoke concentrations, which at times greatly exceeded national safety standards [12]. Prevailing winds largely restricted the dispersion of the smoke plume to the Latrobe Valley SA3 region (which includes Morwell), and that area is considered to have suffered a moderate level of exposure [12]. Further to the east, the SA3 area of Wellington Shire, with a socioeconomic profile similar to that of Latrobe Valley, had little to no exposure to the smoke. Therefore, Wellington Shire was chosen as the comparison area for this analysis. There were 69 primary and secondary schools in the region which were classified into three mine fire exposure groups: Morwell (high exposure), the remainder of the Latrobe Valley (moderate exposure) and Wellington (no/low exposure), see Fig 1.”

Line 151. Some explanation on the interpretation of the ICSEA is needed (higher or lower scores indicate more SE advantage?)

Response: Thank you, this has now been included in the manuscript, lines 183-187. 

“ICSEA is a scale indicating socioeducational advantage of the school based on students’ family background information, which ranges from about 500 (extremely disadvantaged student backgrounds) to about 1300 (extremely advantaged student backgrounds)."

Figure 1 would be benefited if the location of the fire is shown (or another figure so the spatial setting is better contextualised regarding the exposure source). Why areas west of Morwell are not considered? (maybe this links to the need of better explanation of the PM25 modelling)

Response: The fire location is now marked in Fig 1. Other than the SA2s immediately to the west of Morwell (which are included in the Moderate Exposure group in Fig 1), areas further to the west are not considered for two reasons: the prevailing winds kept the smoke plume largely restricted to the Latrobe Valley area, and areas further west of Morwell typically have a higher socioeconomic status. The Latrobe Valley is an area with a relatively low socioeconomic status, which is more comparable to Wellington Shire to the east of the Latrobe Valley. This information was provided in lines 146-155. 

Line 169:

Response: Thank you, this has now been corrected, now line 203.

Line 170. Any previous research using this approach?

Response: No. We haven’t been able to find any similar studies. Almost all studies use individual-level academic scores. There are a few new studies used school-level scores, but only the mean scores were used, hence our argument that this statistical approach is a valuable addition to the field. 

Good inclusion of figure 2

Response: Thank you. 

Statistical analysis is well introduced and supported

Response: Thank you. 

Line 228. Some description of the priors tested in the sensitivity analyses and the main outcomes of excluding cohort RE should be included in the supp. Material

Response: The sensitivity analysis of excluding cohort RE was included in Table S6 in the Supplementary Material. However, the sensitivity analysis using different priors was not because we have tested a few choices, e.g., non-informative priors, weakly informative priors for random effects and non-informative priors for coefficients, and weakly informative priors for all parameters. As a result, it was challenging to include all of these sensitivity analyses. As we have provided source data (simulated), interested users can test different models using the data provided.

Results: Table 1 presents 2 different measures in parentheses -this needs to be better clarified in the table title.

Response: We have changed the notes to better describe the statistics reported. 

“Statistics presented are either the median (IQR: Q1, Q3) or sample size n (%)” 

Line 262. Review these figures

Response: The exact range was -3.68 to -13.41. We approximate by the lower and upper bound, which is -3 and -14. We have now changed the numbers to represent the exact range.

Lines 264-27 data already shown in table 2

Response: We have simplified the sentence as: 

“Table 2 also shows that there were substantial interruption effects post-mine fire in high exposure schools across all academic domains, with estimated mean score reductions of between 10.31 and 11.09 for grammar and punctuation, numeracy, spelling and writing, and 8.34 for reading.”

Paragraph starting in line 290. This answers one of my comments above.

Response: Thank you, change made.

Discussion Line 303. “The results suggest..”

Response: Thank you, change made.

Line 305. Suggest” “the detrimental impact…” (avoid confusion —impacts can also be positive)

Response: Thank you, change made.

Line 326. “The potential links…”

Response: Thank you, change made.

Line 331. “the Hazelwood mine fire smoke”

Response: Thank you, change made.

Line 363. “also” seems redundant here

Response: Thank you, “also” has been deleted.

Line 366. Previous section did not clarify specifically how the analysis address the spatial profile of the impacts (models do not include a spatial term?)

Response: The spatial profile refers to the geographically defined exposure zones. We have included the following text to make this clear. 

“Firstly, it enables the evaluation of the spatial (no/low exposure to high exposure areas) and temporal profile of the impact, which could be used to inform policy and resource allocation in academic settings after disasters.” 

The discussion needs a section to discuss the method used in this study—the strengths section somewhat addresses this but there is not a formal discussion of the specifics of the methods and comparison with other research

Response: We have included a separate section on statistical modelling approach and moved relevant information into this section, see lines 411-439.

The limitations section should address potential ecological bias and how this might affect the results and their interpretation. Also how the limitations already commented in this section might impact the findings and how this could be controlled in further research.

Response: We agree that the ecological bias may impact our findings. Therefore, we have expanded our discussion around this, as follows: 

“Furthermore, the aggregated nature of the data prohibited controlling for individual-level risk factors. While the interrupted time series design was used to allow post-mine fire scores in individual schools to be compared relative to their past, there may be other changes in students’ profiles over time, which may impact our findings. For example, some families may have left the region due to the subsequent closure of the mine and power station. Advanced population-level data linkage methods can be used to explore these issues. However, as noted above, accessing population-level data can be challenging and time-consuming.”

Conclusions: Suggest naming the fire event specifically in the first line

Response: Updated, thank you.

Last line would be better commented in the discussion not in this section

Response: The last sentence was moved to the discussion part on the method.

Response: We have updated the manuscript according to the PLOS ONE's style requirements

2. Please provide additional details regarding participant consent. In the ethics statement in the Methods and online submission information, please ensure that you have specified (1) whether consent was informed and (2) what type you obtained (for instance, written or verbal, and if verbal, how it was documented and witnessed). If your study included minors, state whether you obtained consent from parents or guardians. If the need for consent was waived by the ethics committee, please include this information. If you are reporting a retrospective study of medical records or archived samples, please ensure that you have discussed whether all data were fully anonymized before you accessed them and/or whether the IRB or ethics committee waived the requirement for informed consent. If patients provided informed written consent to have data from their medical records used in research, please include this information.

Response: Only aggregated school-level public data was used therefore individual-level consent was not needed. 

3. Thank you for stating the following financial disclosure: “This work was funded by the Victorian Department of Health and Human Services.” Please state what role the funders took in the study. If the funders had no role, please state: ""The funders had no role in study design, data collection and analysis, decision to publish, or preparation of the manuscript." If this statement is not correct you must amend it as needed. Please include this amended Role of Funder statement in your cover letter; we will change the online submission form on your behalf.

Response: While the overarching design for the wider Hazelwood Health Study was one of the factors assessed by the funders in the original awarding of the research contract, the funders had no role in the specific design, data collection or analysis of this research component. While the funders did not play a role in the decision to publish or the preparation of the manuscript, they were provided with the opportunity to review and approve the manuscript before submission. As part of this review process, the funders made minor suggestions regarding inconsistent language which were corrected. There were no changes made to the analysis and interpretation as a result of this feedback. 

4. Thank you for stating the following in your Competing Interests section: “NO authors have competing interests.” Please complete your Competing Interests on the online submission form to state any Competing Interests. If you have no competing interests, please state ""The authors have declared that no competing interests exist."", as detailed online in our guide for authors at http://journals.plos.org/plosone/s/submit-now This information should be included in your cover letter; we will change the online submission form on your behalf.

Response: please see this information in the cover letter. 

5. We noted in your submission details that a portion of your manuscript may have been presented or published elsewhere. [The manuscript is published on the preprint server https://www.medrxiv.org/content/10.1101/2021.03.28.21254516v2] Please clarify whether this [conference proceeding or publication] was peer-reviewed and formally published. If this work was previously peer-reviewed and published, in the cover letter please provide the reason that this work does not constitute dual publication and should be included in the current manuscript.

Response: This manuscript identified above is a pre-print version of this article which was not peer-reviewed. This information was provided in the original cover letter and we have included this again as requested. 

6. In your Data Availability statement, you have not specified where the minimal data set underlying the results described in your manuscript can be found. PLOS defines a study's minimal data set as the underlying data used to reach the conclusions drawn in the manuscript and any additional data required to replicate the reported study findings in their entirety. All PLOS journals require that the minimal data set be made fully available. For more information about our data policy, please see http://journals.plos.org/plosone/s/data-availability. Upon re-submitting your revised manuscript, please upload your study’s minimal underlying data set as either Supporting Information files or to a stable, public repository and include the relevant URLs, DOIs, or accession numbers within your revised cover letter. For a list of acceptable repositories, please see http://journals.plos.org/plosone/s/data-availability#loc-recommended-repositories. Any potentially identifying patient information must be fully anonymized. Important: If there are ethical or legal restrictions to sharing your data publicly, please explain these restrictions in detail. Please see our guidelines for more information on what we consider unacceptable restrictions to publicly sharing data: http://journals.plos.org/plosone/s/data-availability#loc-unacceptable-data-access-restrictions. Note that it is not acceptable for the authors to be the sole named individuals responsible for ensuring data access. We will update your Data Availability statement to reflect the information you provide in your cover letter.

Response: The data used in this study belonged to a third party, the Australian Curriculum, Assessment and Reporting Authority (ACARA) and access to the data requires permission from the ACARA data custodians. As such, we are unable to make a minimal data set available. Instead, a synthetically generated data set based on the source data for the tutorial part of the paper has been made available at https://doi.org/10.5281/zenodo.6613107

7. We note that you have indicated that data from this study are available upon request. PLOS only allows data to be available upon request if there are legal or ethical restrictions on sharing data publicly. For more information on unacceptable data access restrictions, please see http://journals.plos.org/plosone/s/data-availability#loc-unacceptable-data-access-restrictions. In your revised cover letter, please address the following prompts: a) If there are ethical or legal restrictions on sharing a de-identified data set, please explain them in detail (e.g., data contain potentially sensitive information, data are owned by a third-party organization, etc.) and who has imposed them (e.g., an ethics committee). Please also provide contact information for a data access committee, ethics committee, or other institutional body to which data requests may be sent. b) If there are no restrictions, please upload the minimal anonymized data set necessary to replicate your study findings as either Supporting Information files or to a stable, public repository and provide us with the relevant URLs, DOIs, or accession numbers. For a list of acceptable repositories, please see http://journals.plos.org/plosone/s/data-availability#loc-recommended-repositories.We will update your Data Availability statement on your behalf to reflect the information you provide.

Response: The data used in this study belong to ACARA and access to the data requires permission from the ACARA data custodians. We have obtained the DOI for the release of the synthetically generated data. 

8. Your ethics statement should only appear in the Methods section of your manuscript. If your ethics statement is written in any section besides the Methods, please delete it from any other section.

Response: Thank you, we have moved the ethics statement to the Methods section. 

References 

1. Gibbs L, Nursey J, Cook J, Ireton G, Alkemade N, Roberts M, et al. Delayed Disaster Impacts on Academic Performance of Primary School Children. Child Development. 2019;90(4):1402-12. doi: https://doi.org/10.1007/s12310-016-9175-2. 

2. Reifels L, Bassilios B, Spittal MJ, King K, Fletcher J, Pirkis J. Patterns and Predictors of Primary Mental Health Service Use Following Bushfire and Flood Disasters. Disaster Medicine and Public Health Preparedness. 2015;9(3):275-82. Epub 2015/04/14. doi: 10.1017/dmp.2015.23. 

3. Turner SL, Forbes AB, Karahalios A, Taljaard M, McKenzie JE. Evaluation of statistical methods used in the analysis of interrupted time series studies: a simulation study. BMC Medical Research Methodology. 2021;21(1):181. doi: 10.1186/s12874-021-01364-0. 

4. Turner SL, Karahalios A, Forbes AB, Taljaard M, Grimshaw JM, Cheng AC, et al. Design characteristics and statistical methods used in interrupted time series studies evaluating public health interventions: a review. J Clin Epidemiol. 2020;122:1-11. Epub 20200225. doi: 10.1016/j.jclinepi.2020.02.006. 

5. Tanriver-Ayder E, Faes C, van de Casteele T, McCann SK, Macleod MR. Comparison of commonly used methods in random effects meta-analysis: application to preclinical data in drug discovery research. BMJ Open Science. 2021;5(1):e100074. doi: https://doi.org/10.1136/bmjos-2020-100074. 

6. Berger E, Carroll M, Maybery D, Harrison D. Disaster Impacts on Students and Staff from a Specialist, Trauma-Informed Australian School. Journal of Child & Adolescent Trauma. 2018;11:1-10. doi: https://doi.org/10.1007/s40653-018-0228-6. 

7. McFarlane AC, Policansky SK, Irwin C. A longitudinal study of the psychological morbidity in children due to a natural disaster. Psychological Medicine. 1987;17(3):727-38. Epub 2009/07/09. doi: https://doi.org/10.1017/S0033291700025964. 

8. Silverman WK, La Greca AM. Children experiencing disasters: Definitions, reactions, and predictors of outcomes. 2002:11-33. doi: 10.1037/10454-001. 

9. Seddighi H, Salmani I, Javadi MH, Seddighi S. Child Abuse in Natural Disasters and Conflicts: A Systematic Review. Trauma, Violence, & Abuse. 2019;22(1):176-85. doi: 10.1177/1524838019835973. 

10. Griffith AK. Parental Burnout and Child Maltreatment During the COVID-19 Pandemic. Journal of Family Violence. 2022;37(5):725-31. doi: 10.1007/s10896-020-00172-2. 

11. ABS. Australian Statistical Geography Standard (ASGS): Australian Bureau of Statistics (ABS); 2011. Available from: https://www.abs.gov.au/websitedbs/d3310114.nsf/home/australian+statistical+geography+standard+(asgs).

12. Luhar AK, Emmerson KM, Reisen F, Williamson GJ, Cope ME. Modelling smoke distribution in the vicinity of a large and prolonged fire from an open-cut coal mine. Atmospheric Environment. 2020;229:117471. doi: https://doi.org/10.1016/j.atmosenv.2020.117471.

---

## [Decision Letter · Decision Letter 1]

14 Sep 2022

PONE-D-21-37171R1Evaluating the impact of Hazelwood mine fire event on students’ educational development with Bayesian interrupted time-series hierarchical meta-regressionPLOS ONE

Dear Dr. Gao

Thank you for submitting your manuscript to PLOS ONE. After careful consideration, we feel that it has merit but does not fully meet PLOS ONE’s publication criteria as it currently stands. Therefore, we invite you to submit a revised version of the manuscript that addresses the points raised during the review process.

We look forward to receiving your revised manuscript.

Kind regards,

Soham Bandyopadhyay

Academic Editor

PLOS ONE

Reviewers' comments:

Reviewer's Responses to Questions

**Comments to the Author**

2. Is the manuscript technically sound, and do the data support the conclusions?

Reviewer #1: Partly

Reviewer #2: Partly

3. Has the statistical analysis been performed appropriately and rigorously? 

Reviewer #1: Yes

Reviewer #2: Yes

4. Have the authors made all data underlying the findings in their manuscript fully available?

Reviewer #1: Yes

Reviewer #2: Yes

5. Is the manuscript presented in an intelligible fashion and written in standard English?

Reviewer #1: Yes

Reviewer #2: Yes

6. Review Comments to the Author

Reviewer #1: The authors have rightly addressed most of my previous comments in the first version of the manuscript. This allowed to have a better appreciation of the analysis and discussion from which some further comments and suggestions should be addressed before acceptance for publication. Please see specific comments for each section below.

Abstract

Line 29. Maybe you meant “publicly available” rather than easily?

Introduction

Lines 59-61. Maybe “environmental disasters” is a better context than “climate disasters” -or a better contextualisation on how this industrial disaster matches a climate event is needed. This would align with the citations used as some relate to natural disasters.

Lines 70, 79. Consider the above comment.

Lines 100-101. This links to a lack of research on health impacts of the coal mining industry in Australia, as health aspects are quite related to cognitive development and performance -this can be addressed in the discussion.

Line 110. Suggest to add specific context of populations affected (i.e., children in the proximity, etc..)

Methods

Line 175. Not sure these coefficients are “comparable” to individual level analyses.

Line 55. Much recent citations can back this statement

Line 206. How does this equation link to the model shown in line 192 (I don’t remember if this was asked in the first review -maybe some of the explanation from lines 207-216 can be better introduced with equations linked to the regression model).

Results

Line 267. Maybe you meant “i.e.,,” rather than “e.g.,”

Discussion

Line 313. Remove “some”

Lines 337-338. The authors should expand more on poor impact assessment in the coal mining sector. Very few but sill some important studies relevant to Australia would support this (consider: doi.org/ 10.1093/heapro/daz032 and doi.org/ 10.1515/reveh-2019-0033)

Lines 346-348. From this line you should reconsider the following statement in the first parag of the discussion […] the …“mine fire had a major detrimental impact on academic performance”

Section “comparison with other studies”. I suggest the authors also address the differences in impacts from different “types” of disaster. Natural disasters most probably have a different reaction/effect at the community level than disaster in industries with big economic involvement -as there is some duty of care related to the lack of prevision (consider the impact assessment suggestions above).

Line 377. Not sure if you are refereeing to a specific context. I identified at least 2 studies that would match this worldwide.

Line 382. I have read the response to a previous comment in this line, but I still think the analysis does not really evaluate the spatial profile of the impacts since the model does not incorporate a spatial term or a spatial structure specification.

The Limitations section needs to address specifically the potential risk of ecological bias. The impact can be significant (potential spurious associations due to analysis at the group rather than individual level) and the authors need to acknowledge this limitation. This also links to how the conclusions are presented. An ecological analysis cannot provide evidence of causality (rather a statistical evidence, or conclusions circumscribed to the context of the analysis).

Reviewer #2: The project is based on an excellent idea: demonstrating how

readily-available administrative data, when analysed in a sensible

way, can provide insight into the effects of local disasters. As the authors state, an analysis of the Hazelwood mine fire could form a template for analyses of disasters elsewhere (where there is good admin data available).

The basic design of the analysis, with two types of control groups plus before-after comparisons, is nice, in that it is easy to understand and has a good chance of isolating the actual effects of the disaster. The authors have done a good job of responding to comments from an earlier reviewer. In my opinion, however, a bit more work is required to bring the modelling itself, plus the description of the modelling, to the point where it would form a good template for analyses of other disasters. My reservations are as follows.

SHRINKAGE

One of the fundamental motivations for using Bayesian hierarchical models is to stabilise estimates, and avoid coefficient estimates that merely reflect random variation. See, for instance, https://www.tandfonline.com/doi/abs/10.1080/19345747.2011.618213

The typical mechanism for achieving shrinkage is to put priors on variance terms that favour values near zero - see, for instance, https://github.com/stan-dev/stan/wiki/Prior-Choice-Recommendations

Some form of shrinkage arguably should be a default, and I would expect it to be included in a template for other analyses. However, judging by the description on pp11-12, the models in the paper do not take advantage of this possibility. Perhaps the authors could (1) explain why they did not build shrinkage into their analysis, or (2) amend the model?

ALGEBRAIC DESCRIPTION OF MODEL

I found the algebraic description of the model on pp10-11 difficult to follow. What is the 'e' subscript? Given that the outcomes are time indexed, it seems like there should be a time subscript, but there isn't one. It seems like the time subscript may be included implicitly in beta and X, but this is confusing, and I'm not sure it's quite correct.

(I looked for the Stan code, to clarify what was going on, but the supplementary materials were not included with the submission - which may be a PLOS problem??)

A more minor point - rather than beta X_{s,g}, where X is a matrix, you probably want x_{s,g} beta, where x_{s,g} is a row vector and beta is a column vector.

IMPORTANCE OF CONTROL GROUPS?

The Abstract and Discussion emphasise the use of admin data, before-after comparisons, and Bayesian modelling. Arguably, the use of appropriate control groups is also an important part of the general template. The results would have been much less convincing if only values for the directly affected schools were shown. Perhaps this requires more emphasis? (And, if so, some thought about problems with using the design to study the effect of COVID on schooling.)

MULTIPLE OUTCOMES

One important feature of the analysis is that it uses multiple outcomes. This creates some challenging problems of interpretation and inference in cases where different outcomes appear to be affected differently. There is not consensus on the best approach to take to multiple outcomes - see, for instance, https://statmodeling.stat.columbia.edu/2022/05/18/doing-mister-p-with-multiple-outcomes/

But maybe it deserves a bit more discussion, and the approach of the paper (separate models for each outcome) needs to be mentioned and defended?

7. PLOS authors have the option to publish the peer review history of their article (what does this mean?). If published, this will include your full peer review and any attached files.

Reviewer #1: No

Reviewer #2: **Yes: **John Bryant

---

## [Author Response · Author response to Decision Letter 1]

5 Oct 2022

Responses to Reviewers

Our sincere thanks to the reviewers for providing such detailed and helpful comments on our manuscript. We have responded to each comment below and highlighted relevant changes in the revised manuscript. 

Reviewer #1: 

The authors have rightly addressed most of my previous comments in the first version of the manuscript. This allowed us to have a better appreciation of the analysis and discussion from which some further comments and suggestions should be addressed before acceptance for publication. Please see specific comments for each section below.

Response: Thank you for these positive comments. We have addressed each of the reviewer’s comments in relation to the paper below.

Line 29. Maybe you meant “publicly available” rather than easily?

Response: Data for individual schools is available online (https://www.myschool.edu.au/). To obtain all data, researchers can apply for data access via the Australian Curriculum Assessment and Reporting Authority (ACARA), which assesses research applications on the basis of merit. Therefore, we feel that “easily accessible” is still slightly better in describing the accessibility of the data. 

Lines 59-61. Maybe “environmental disasters” is a better context than “climate disasters” -or a better contextualisation on how this industrial disaster matches a climate event is needed. This would align with the citations used as some relate to natural disasters. Lines 70, 79. Consider the above comment.

Response: We agree with the reviewer’s perspective that “environmental disasters” is a broader and better term to use in our paper. We have updated accordingly throughout the document. 

Lines 100-101. This links to a lack of research on health impacts of the coal mining industry in Australia, as health aspects are quite related to cognitive development and performance -this can be addressed in the discussion.

Response: Thanks for pointing this out. We have added additional discussions in lines 355-361. 

“Fire incidents are common in coal mines [70] and, all too often, mining project impact assessments adopt a narrow range of health and wellbeing domains that omit consideration of potential social consequences [71]. Accordingly, impacts that events such as the Hazelwood mine fire can have on academic outcomes among school-aged young people warrant consideration within mine project planning and, more generally, within disaster risk reduction planning and education department policies.”

Line 110. Suggest to add specific context of populations affected (i.e., children in the proximity, etc..)

Response: The indicator of geospatial location was included (lines 101-102; inserted text in bold below) and we have also referenced the air pollution study for further details. 

“Whilst the flames themselves did not directly threaten homes or lives, heavy smoke concentrations throughout the six-week period (particularly in the nearby town of Morwell and the wider Latrobe valley area) [42] resulted in increased mortality, physical ill-health and psychological distress in the local community [35, 43-47].”

Methods Line 175. Not sure these coefficients are “comparable” to individual level analyses.

Response: The results from Bayesian models using school-level distributions should obtain similar results to modelling outcomes based on individual students’ records, as long as the assumptions of the Bayesian models can be met. While there is a clear benefit in using individual-level data because it allows individual-level factors (e.g., age, gender, adverse childhood events, parent educational status) to be considered, accessing such data can be extremely challenging. Although we took some of these factors into consideration in the model (e.g., school type, percentages of female students, area-based socioeconomic status), not including individual-level data may still have an impact on the results. In this study, it is unlikely that some of these factors caused differences in exposure (i.e. adverse childhood events will not cause a higher or lower mine fire exposure level). However, we agree that it is important to make the claim clear. We have modified the text to be more specific, see line 183.

“This approach directly models distributions of students’ performance within schools using mean scores and standard errors, which provides unbiased estimates that are broadly comparable with the model using individual-level records (not considering individual-level confounding factors).”

Methods: Line 55. Much recent citations can back this statement

Response: Thank you. We have included the updated CRED report and the most recent IPCC report.

“The frequency and severity of disasters such as wildfires have increased dramatically in recent years within the context of climate change [2-4].”

“2. Xu R, Yu P, Abramson MJ, Johnston FH, Samet JM, Bell ML, et al. Wildfires, Global Climate Change, and Human Health. New England Journal of Medicine. 2020;383(22):2173-81. doi: https://doi.org/10.1056/NEJMsr2028985 . 

3. IPCC. Summary for Policymakers. In: Climate Change 2021: The Physical Science Basis. Contribution of Working Group I to the Sixth Assessment Report of the Intergovernmental Panel on Climate Change. 2022. Available from: https://www.ipcc.ch/report/ar6/wg1/#SPM . 

4. CRED. The Non-Covid year in disasters: Global trends and perspectives. Centre for Research on the Epidemiology of Disasters (CRED), UN Office for Disaster Risk Reduction., 2021. Available from: http://hdl.handle.net/2078.1/245181.”

Methods: Line 206. How does this equation link to the model shown in line 192 (I don’t remember if this was asked in the first review -maybe some of the explanations from lines 207-216 can be better introduced with equations linked to the regression model).

Response: It is slightly unclear to us whether “Line 206” refers to the track-changed document or no track-changed document. The question makes sense in both these two places. 

If “Line 206” refers to the equation (1) in the document without track-change, we included additional information that equation (2) is the expansion of the interrupted time series terms in equation (1) in lines 215-216:

“The interrupted time series terms in X_(s,g) β in Equation (1) are summarised as follows”

However, “Line 206” can also refer to “Interrupted time series models using only means with linear regression instead of meta-regression will be inefficient (significant loss of statistical power) and biased towards smaller schools (large schools with more students should be given more weights)” in the document with track-change. If this is the case, we have included additional information regarding weights in lines 183-187.

“Interrupted time series models using only means with linear regression instead of meta-regression will be inefficient (significant loss of statistical power) and biased towards smaller schools (large schools with more students should be given more higher weights, e.g., using regression models weighted by the number of students or meta-regressions with inverse variance as weights).” 

Results: Line 267. Maybe you meant “i.e.,,” rather than “e.g.,”

Response: This is now corrected (see line 285). Thank you. 

Discussion: Line 313. Remove “some”

Response: This has been deleted (see line 331).

Discussion: Lines 337-338. The authors should expand more on poor impact assessment in the coal mining sector. Very few but still some important studies relevant to Australia would support this (consider: doi.org/ 10.1093/heapro/daz032 and doi.org/ 10.1515/reveh-2019-0033), see lines 355-361.

Response: We have added an assertion within the discussion section in relation to the reviewers’ suggestion here, and have made use of one of the references forwarded. 

“Fire incidents are common in coal mines [70] and, all too often, mining project impact assessments adopt a narrow range of health and wellbeing domains that omit consideration of potential social consequences [71]. Accordingly, impacts that events such as the Hazelwood mine fire can have on academic outcomes among school-aged young people warrant consideration within mine project planning and, more generally, within disaster risk reduction planning and education department policies.”

Discussion: Lines 346-348. From this line you should reconsider the following statement in the first parag of the discussion […] the …“mine fire had a major detrimental impact on academic performance”

Response: Thank you for pointing this out. We use “delineate the relative contributions” to refer to the fact that there are a few possible mechanisms of how mine fire exposure may contribute to the overall detrimental impact. Therefore, further investigation is needed to understand which mechanism is more important and to develop targeted interventions. We have modified the text to make this clear, see lines 365-376.

“Although our study has shown a clear impact of the mine fire on academic performance, the model cannot delineate between the potential underlying mechanisms related to the exposure, such as psychological trauma, disruption to schooling, or air pollution exposure. Accordingly, further investigation is needed to understand the relative contributions of different mechanisms by which disaster exposure impacts academic performance, which could inform the development of targeted interventions.” 

Discussion: Section “comparison with other studies”. I suggest the authors also address the differences in impacts from different “types” of disaster. Natural disasters most probably have a different reaction/effect at the community level than disaster in industries with big economic involvement -as there is some duty of care related to the lack of prevision (consider the impact assessment suggestions above).

Response: While there is a dearth of literature regarding whether academic performance is differentially impacted by whether disasters are natural or human-caused, there is evidence to suggest that other health outcomes such as the lifetime risk of psychiatric disorders are more associated with human-caused events [1]. However, it is slightly difficult to classify Hazelwood mine fire in the spectrum of “natural” to “man-made” disasters. The mine fire was initiated by a bushfire. However, a significant issue was the apparent lack of preparedness for a mine fire, with the state-government commissioned Hazelwood Mine Fire Inquiry noting that “all of the factors contributing to the ignition and spread of the fire were foreseeable. Yet it appears they were not foreseen” [2]. This failure to take appropriate measures to prevent a likely fire meant that the community likely saw the mine fire as having both natural and human causes. In line with the reviewers' suggestion, we have expanded the discussion at lines 404-412, including directions for future research.

“Longer-term social outcomes after man-made or industrial disasters have the potential to be worse than in cases of natural disasters [85], however, it is at present unclear whether this also applies to academic outcomes. Furthermore, while there is little research available for assessing the relative impacts of different types of disasters on academic outcomes, it is reasonable to anticipate that more severe disasters will have a potentially greater impact on academic outcomes. For instance. if school infrastructure was destroyed, or if students were exposed to the trauma of having lost family or classmates in the event, then academic consequences might be expected to be worse than in the case of the Hazelwood mine fire. Accordingly, seeking to compare how different types of disaster shape academic outcomes would be a useful direction for future research.” 

Discussion: Line 377. Not sure if you are refereeing to a specific context. I identified at least 2 studies that would match this worldwide.

Response: We agree with the reviewer that we could be more specific in this assertion. We are not aware of any studies having used aggregated school level-data in this Bayesian multi-level meta-regression framework. A very simple illustrative case was used in Andrew Gelman’s book, Bayesian Data Analysis, which inspired this study. Please feel free to let us know if we missed anything. 

“To the best of our knowledge, this is the first study using a Bayesian meta-regression method to evaluate aggregated school-level academic outcomes.” 

Discussion: Line 382. I have read the response to a previous comment in this line, but I still think the analysis does not really evaluate the spatial profile of the impacts since the model does not incorporate a spatial term or a spatial structure specification.

Response: We agree that the model does not include a strictly defined spatial term. Our “spatial” term refers to geographical areas of low, medium and high exposure levels (see lines 419-420). 

“Firstly, it enables the evaluation of the spatial (geographical regions of no/low exposure to high exposure areas)”

Although we can estimate the continuous spatial term (e.g., distance to the mine) and include a spatial and temporal interaction, it is not possible to interpret the results (there is both the direct interruption effect and the change of slope effect). We feel that it would be suitable for readers to consider that the method can model differential effects based on geographical location.

Limitations: The Limitations section needs to address specifically the potential risk of ecological bias. The impact can be significant (potential spurious associations due to analysis at the group rather than individual level) and the authors need to acknowledge this limitation. This also links to how the conclusions are presented. An ecological analysis cannot provide evidence of causality (rather a statistical evidence, or conclusions circumscribed to the context of the analysis).

Response: Thank you, we have now addressed this in the discussion. The interrupted time-series design aims at evaluating changes in population trends associated with an event, which is less likely to be confounded by individual-level factors except when the individual factor is associated with the event itself [3]. We have also included a control group to reduce the impact of time-varying confounders. In lines 422-426, we have added the benefits of including a control group in the analysis. 

“The inclusion of an appropriate control group of schools from areas with similar socioeconomic circumstances, but not exposed to smoke during the fire, countered the problem of potential time-varying confounders (e.g., changes in tests, education policy or other major interruptions) and enabled the impact of the mine fire on academic progress in the region to be clearly isolated [91]”

In lines 474-475 we have included ecological bias as a potential limitation.

“Furthermore, the aggregated nature of the data precluded controlling for individual-level risk factors and, hence, there is potential for ecological bias within the study.”

Reviewer #2:

The project is based on an excellent idea: demonstrating how readily-available administrative data, when analysed in a sensible way, can provide insight into the effects of local disasters. As the authors state, an analysis of the Hazelwood mine fire could form a template for analyses of disasters elsewhere (where there is good admin data available). The basic design of the analysis, with two types of control groups plus before-after comparisons, is nice, in that it is easy to understand and has a good chance of isolating the actual effects of the disaster. The authors have done a good job of responding to comments from an earlier reviewer. In my opinion, however, a bit more work is required to bring the modelling itself, plus the description of the modelling, to the point where it would form a good template for analyses of other disasters. My reservations are as follows.

Response: Thank you for these positive comments. We have addressed the comments in relation to the paper below.

One of the fundamental motivations for using Bayesian hierarchical models is to stabilise estimates, and avoid coefficient estimates that merely reflect random variation. See, for instance, https://www.tandfonline.com/doi/abs/10.1080/19345747.2011.618213

Response: Fantastic suggestion. Gelman’s paper is a great reference. We have included additional discussion in lines 445-448.

“However, fitting models with the Bayesian approach has additional advantages such as more stable and generalisable estimates, integration of existing knowledge using prior distributions, flexible compressions using posterior distributions and avoiding convergence and estimation problems of maximum likelihood methods [61,93].”

The typical mechanism for achieving shrinkage is to put priors on variance terms that favour values near zero - see, for instance, https://github.com/stan-dev/stan/wiki/Prior-Choice-Recommendations. Some form of shrinkage arguably should be a default, and I would expect it to be included in a template for other analyses. However, judging by the description on pp11-12, the models in the paper do not take advantage of this possibility. Perhaps the authors could (1) explain why they did not build shrinkage into their analysis, or (2) amend the model?

Response: We believe the reviewer is referring to partial pooling of the random intercepts. Our two distributions for the random effects - one for the school random intercepts, another for the cohort random intercepts - each assume a single standard deviation term, on which we place a weakly informative prior. This common SD ensures some level of partial pooling across schools and cohorts. Our truncated normal prior for the random effects SD did use a mean of 10 rather than 0, which may be what the reviewer is referring to in their comment. The mean of 10 was chosen instead of 0 because there were slightly fewer high-performance schools and we did not want to underestimate this variation. However, the weekly prior distribution had very little impact on our results. See examples below:

Using the mean of 0 in prior distributions in the tutorial code.

Using the mean of 10 in prior distributions in the tutorial code.

We have included this in lines 238-240.

“A mean of 0 is normally used for prior distributions of the random effects to take more advantage of the partial pooling effect of the multi-level model [61]. In our case, the mean of 10 was chosen as the high-performance schools were slightly underrepresented in the area and variations across all regional Victorian schools were used (based on a preliminary exploratory evaluation).”

The reviewer has also reminded us to mention the possible usage of shrinkage priors, please see our updates in lines 448-451.

“The choice of prior distributions did not impact our results, however, when there is high model complexity (e.g., higher numbers of potential confounding factors), penalising Bayesian models can be used with shrinkage prior distributions [94, 95].”

ALGEBRAIC DESCRIPTION OF MODEL: I found the algebraic description of the model on pp10-11 difficult to follow. What is the 'e' subscript? 

Response: The 'e' subscript represents exposure. These are six terms associated with exposure, which were listed separately to explain how to interpret mine fire exposure-related effects. To increase the readability of these notations, we have modified as follows:

 β_(m,prior) and β_(h,prior): prior mine fire differences for moderate and high exposure schools

 β_(m,int) and β_(h,int): mine-fire interruption effects 

 β_(m,trend) and β_(h,trend): post-mine fire trend differences

See lines 214-230 for all the updates. 

Given that the outcomes are time indexed, it seems like there should be a time subscript, but there isn't one. It seems like the time subscript may be included implicitly in beta and X, but this is confusing, and I'm not sure it's quite correct. 

The model includes two temporal variables. One is the year (continuous variable) as a long-term trend, which is described as a confounder. Another temporal term is the T_post - post-mine fire time variable (0 for 2008-2014, 1 for 2015, 2 for 2016 etc) which was included in the exposure time interaction term in the second equation, which is a standard interrupted time series parameterisation [4]. We have included the long-term trend term (T) in Equation (2) to make this clear. 

I looked for the Stan code, to clarify what was going on, but the supplementary materials were not included with the submission - which may be a PLOS problem??

PLOS one places the Supplementarily Material at the end of the full-text (before the track-changed document). The data and Stan code can also be found on the data repository Zenodo: http://doi.org/10.5281/zenodo.6613107.

A more minor point - rather than beta X_{s,g}, where X is a matrix, you probably want x_{s,g} beta, where x_{s,g} is a row vector and beta is a column vector.

Response: Yes, it should be X_(s,g) β, this is corrected in Equation (1) 

IMPORTANCE OF CONTROL GROUPS? The Abstract and Discussion emphasise the use of admin data, before-after comparisons, and Bayesian modelling. Arguably, the use of appropriate control groups is also an important part of the general template. The results would have been much less convincing if only values for the directly affected schools were shown. Perhaps this requires more emphasis? (And, if so, some thought about problems with using the design to study the effect of COVID on schooling.)

Response: We agree with this suggestion. In this particular case, schools in SA3 areas with similar socioeconomic profiles but not impacted by the mine-fire smoke were chosen for the control group. As suggested, the control group in the study is valuable for ensuring that the changes observed were not due to other time-varying factors such as changes in assessment content, funding policy, other major interruptions etc. The control group improves the robustness of the finding and we have included additional discussion in lines 422-426.

“The inclusion of an appropriate control group of schools from areas with similar socioeconomic circumstances, but not exposed to smoke during the fire, countered the problem of potential time-varying confounders (e.g., changes in tests, education policy or other major interruptions) and enabled the impact of the mine fire on academic progress in the region to be clearly isolated [91].”

MULTIPLE OUTCOMES: One important feature of the analysis is that it uses multiple outcomes. This creates some challenging problems of interpretation and inference in cases where different outcomes appear to be affected differently. There is not consensus on the best approach to take to multiple outcomes - see, for instance, https://statmodeling.stat.columbia.edu/2022/05/18/doing-mister-p-with-multiple-outcomes/. But maybe it deserves a bit more discussion, and the approach of the paper (separate models for each outcome) needs to be mentioned and defended?

Response: This is definitely worth additional discussion. From an educational psychology point of view, it would be remiss to combine all domain scores into a single score, as general developmental trends are different across domains. It would also be presumptuous to expect the impacts would be the same across all domains, as some domains might be more vulnerable to immediate changes and interruptions. These are the practical reasons behind modelling the outcomes separately. As the reviewer suggested there is currently no consensus on the best approach. We have included the following additional discussion in the limitation part and highlighted that when the assumption or interest is to understand a consistent impact, then the single multi-level model can be used, see lines 466-473. 

“Outcomes across multiple learning domains were compared which can introduce challenges for interpretation. In this study, it is likely that interruptions caused by the mine fire to school routines were distributed similarly across all learning domains; however, the level of impact of this disruption on students’ subsequent progression within each learning domain may differ. Therefore, each domain was evaluated separately to investigate for any differing impacts and trends between domains. Alternatively, if there was an assumption or interest in evaluating a consistent impact across learning domains, all could be included in one multilevel model with an additional nested level [61].”

References 

1. Reifels L, Mills K, Dückers MLA, O'Donnell ML. Psychiatric epidemiology and disaster exposure in Australia. Epidemiology and Psychiatric Sciences. 2019;28(3):310-20. Epub 2017/09/27. doi: 10.1017/S2045796017000531. 

2. Teague B, Catford J, Petering S. Hazelwood Mine Fire Inquiry Report. Vol. 1. . Vic Government Printers, 2014. Available from: https://www.parliament.vic.gov.au/file_uploads/8101_HAZ_Hazelwood_Mine_Inquiry_Report_BOOK_LR_f5Bp6wNh.pdf

3. Penfold RB, Zhang F. Use of interrupted time series analysis in evaluating health care quality improvements. Acad Pediatr. 2013;13(6 Suppl):S38-44. doi: https://doi.org/10.1016/j.acap.2013.08.002. 

4. Penfold RB, Zhang F. Use of Interrupted Time Series Analysis in Evaluating Health Care Quality Improvements. Academic Pediatrics. 2013;13(6, Supplement):S38-S44. doi: https://doi.org/10.1016/j.acap.2013.08.002.

---

## [Decision Letter · Decision Letter 2]

13 Jan 2023

PONE-D-21-37171R2Evaluating the impact of Hazelwood mine fire event on students’ educational development with Bayesian interrupted time-series hierarchical meta-regressionPLOS ONE

Dear Dr. Gao,

Thank you for submitting your manuscript to PLOS ONE. After careful consideration, we feel that it has merit but does not fully meet PLOS ONE’s publication criteria as it currently stands. Therefore, we invite you to submit a revised version of the manuscript that addresses the points raised during the review process.

Thank you for your patience during this extended review process.Please address Reviewer #1 comment #1 by providing further justification to support the indicated statement or revising it as requested.You may wish to take this opportunity to address the other recommendations from both reviewers.==============================

We look forward to receiving your revised manuscript.

Kind regards,

Toby Mansell, PhD, MBiostat

Academic Editor

PLOS ONE

Journal Requirements:

Reviewers' comments:

Reviewer's Responses to Questions

**Comments to the Author**

1. If the authors have adequately addressed your comments raised in a previous round of review and you feel that this manuscript is now acceptable for publication, you may indicate that here to bypass the “Comments to the Author” section, enter your conflict of interest statement in the “Confidential to Editor” section, and submit your "Accept" recommendation.

Reviewer #1: All comments have been addressed

Reviewer #2: All comments have been addressed

2. Is the manuscript technically sound, and do the data support the conclusions?

Reviewer #1: Yes

Reviewer #2: Yes

3. Has the statistical analysis been performed appropriately and rigorously? 

Reviewer #1: Yes

Reviewer #2: Yes

4. Have the authors made all data underlying the findings in their manuscript fully available?

Reviewer #1: Yes

Reviewer #2: Yes

5. Is the manuscript presented in an intelligible fashion and written in standard English?

Reviewer #1: Yes

Reviewer #2: Yes

6. Review Comments to the Author

Reviewer #1: I appreciate the authors have considered my previous comments. From reviewing this current version of the manuscript, there are just two additional aspects that still need to be addressed before acceptance for publication:

1) In the response to the comparability to individual level analyses, I don’t agree that “the school-level distributions should obtain similar results to modelling outcomes based on individual students’ records”. The use of school level data implies an ecological model and I understand there is not a statistical approach to overcome this limitation. This was previously discussed in the first review. Reconsider to update the statement in line 183 to reflect accurately on this.

2) I think the discussion still needs more support on local studies on the lack of quality impact assessment in the Australian coal mining sector. Research on the impacts of coal mining in the country is scarce and it is of value to highlight previous relevant research. Consider again the references commented in the first review.

Reviewer #2: I am satisfied that the authors seriously considered all the questions that I raised in my review. I do not always agree with the authors' choices. I continue to find the use of the a truncated normal with mu=10 strange, in that it actually pulls the posterior away from zero in cases where the observed standard deviation is below 10. However, reasonable people can disagree on these things, and the authors note that the results are not sensitive to the choice of prior.

One minor suggestion: in the notation on 214-220, perhaps, in a Bayesian context, the terms 'prior' and 'post' are not ideal, as they invite confusion with prior and posterior distributions. Better to use 'before' and 'after'?

7. PLOS authors have the option to publish the peer review history of their article (what does this mean?). If published, this will include your full peer review and any attached files.

Reviewer #1: No

Reviewer #2: **Yes: **John Bryant

---

## [Author Response · Author response to Decision Letter 2]

23 Jan 2023

Responses to Reviewers

Our sincere thanks to the reviewers for providing such detailed and helpful comments on our manuscript. We have responded to each comment below and highlighted relevant changes in the revised manuscript. 

Reviewer #1: 

I appreciate the authors have considered my previous comments. From reviewing this current version of the manuscript, there are just two additional aspects that still need to be addressed before acceptance for publication:

Response: Thank you for these positive comments. We have addressed the comments in relation to the paper below.

In the response to the comparability to individual level analyses, I don’t agree that “the school-level distributions should obtain similar results to modelling outcomes based on individual students’ records”. The use of school level data implies an ecological model and I understand there is not a statistical approach to overcome this limitation. This was previously discussed in the first review. Reconsider to update the statement in line 183 to reflect accurately on this.

Response: Thanks for the suggestion. We may not make our argument clear enough here. We did not intend to infer that ecological models are the same as individual-level models (ecological bias is well-known and were discussed in the limitation part). Here we are refereeing to the fact that the meta-analysis, which considered both the means and SEs, is unbiased compared with including individual-level models without confounder adjustment. If only mean scores were used, then the results will be biased (smaller schools with wider distribution will be given more weights than there should be). We have modified the discussion to reflect this more clearly as follows:

“This approach directly models distributions of students’ performance within schools using mean scores and standard errors, which follows the meta-analysis approach to ensure unbiased estimates that are comparable with the individual-level model without confounder adjustment (assuming there is no ecological bias).”

I think the discussion still needs more support on local studies on the lack of quality impact assessment in the Australian coal mining sector. Research on the impacts of coal mining in the country is scarce and it is of value to highlight previous relevant research. Consider again the references commented in the first review.

Response: We have included the recommended reference and discussion in the last review submission. We have modified this section according to the reviewer’s suggestion. 

“Fire incidents are common in coal mines [70] and, all too often, mining project impact assessments adopt a narrow range of health and wellbeing domains that omit consideration of epidemiological evidence and potential social consequences [71, 72]. Accordingly, the impacts that events such as the Hazelwood mine fire can have on academic outcomes among school-aged young people warrant consideration within mine project planning and impact assessment, more generally, within disaster risk reduction planning and education department policies.”

Reviewer #2:

I am satisfied that the authors seriously considered all the questions that I raised in my review. I do not always agree with the authors' choices. I continue to find the use of a truncated normal with mu=10 strange, in that it actually pulls the posterior away from zero in cases where the observed standard deviation is below 10. However, reasonable people can disagree on these things, and the authors note that the results are not sensitive to the choice of prior.

Response: Thank you for these positive comments. Our choice is perhaps more impacted by our prior beliefs when working with schools in the area. The schools in the region would likely be at least as diverse as if not more diverse than the rest of the regional areas. As the reviewer suggested It is more reassuring that prior distributions had very little impact on results. We have addressed the remaining comment below.

One minor suggestion: in the notation on 214-220, perhaps, in a Bayesian context, the terms 'prior' and 'post' are not ideal, as they invite confusion with prior and posterior distributions. Better to use 'before' and 'after'?

Response: We agree that the term “prior” can be easily confused with “prior distribution”. We have also noticed that we have used “pre” and “prior” interchangeably, which may also contribute to this confusion. It is slightly awkward to refer to the “post-mine fire trend” as “the trend after the mine fire”, so we decided to keep this and change “prior-” and “post-” mine fire to “pre-” and “post-” mine fire. Hopefully, this can avoid confusion as “pre and post-disaster” are commonly used terms.

---

## [Editor Report · Decision Letter 3]

30 Jan 2023

Evaluating the impact of Hazelwood mine fire event on students’ educational development with Bayesian interrupted time-series hierarchical meta-regression

PONE-D-21-37171R3

Dear Dr. Gao,

We’re pleased to inform you that your manuscript has been judged scientifically suitable for publication and will be formally accepted for publication once it meets all outstanding technical requirements.

Kind regards,

Toby Mansell, PhD, MBiostat

Academic Editor

PLOS ONE

---

## [Editor Report · Acceptance letter]

2 Feb 2023

PONE-D-21-37171R3 

Evaluating the impact of Hazelwood mine fire event on students’ educational development with Bayesian interrupted time-series hierarchical meta-regression 

Dear Dr. Gao:

I'm pleased to inform you that your manuscript has been deemed suitable for publication in PLOS ONE. Congratulations! Your manuscript is now with our production department. 

Kind regards, 

on behalf of

Dr. Toby Edward Mansell 

Academic Editor

PLOS ONE